# NULL COUNTERFACTUAL FACTOR INTERACTIONS FOR GOAL-CONDITIONED REINFORCEMENT LEARNING

**Caleb Chuck**[1*]**, Fan Feng**[2,3*]**, Carl Qi**[1]**, Chang Shi**[1]**, Siddhant Agarwal**[1]**,
Amy Zhang**[1]**, Scott Niekum**[4]
[1] The University of Texas at Austin [2] University of California San Diego
[3] MBZUAI [4] University of Massachusetts Amherst

## ABSTRACT

Hindsight relabeling is a powerful tool for overcoming sparsity in goal-conditioned reinforcement learning (GCRL), especially in certain domains such as navigation and locomotion. However, hindsight relabeling can struggle in object-centric domains. For example, suppose that the goal space consists of a robotic arm pushing a particular target block to a goal location. In this case, hindsight relabeling will give high rewards to any trajectory that does not interact with the block. However, these behaviors are only useful when the object is already at the goal—an extremely rare case in practice. A dataset dominated by these kinds of trajectories can complicate learning and lead to failures. In object-centric domains, one key intuition is that meaningful trajectories are often characterized by object-object interactions such as pushing the block with the gripper. To leverage this intuition, we introduce Hindsight Relabeling using Interactions (HInt), which combines interactions with hindsight relabeling to improve the sample efficiency of downstream RL. However, interactions do not have a consensus statistical definition that is tractable for downstream GCRL. Therefore, we propose a definition of interactions based on the concept of *null counterfactual*: a cause object is interacting with a target object if, in a world where the cause object did not exist, the target object would have different transition dynamics. We leverage this definition to infer interactions in Null Counterfactual Interaction Inference (NCII), which uses a "nulling" operation with a learned model to simulate absences and infer interactions. We demonstrate that NCII is able to achieve significantly improved interaction inference accuracy in both simple linear dynamics domains and dynamic robotic domains in Robosuite, Robot Air Hockey, and Franka Kitchen. Furthermore, we demonstrate that HInt improves sample efficiency by up to $4\times$ in these domains as goal-conditioned tasks.

## 1 INTRODUCTION

Reinforcement Learning (RL) has made great strides when applied to specific tasks with clear, well-designed rewards (Silver et al., 2018; Wurman et al., 2022; Trinh et al., 2024) but learning generalist policies remains an open problem. This is because a vanilla RL is formulated as the optimization of a single reward function. Goal-conditioned Reinforcement Learning (GCRL) offers a powerful mechanism of generalization by conditioning the learned policy on a variety of goals. These goals can be parameterized by a particular setting of an object, such as hitting a puck to a goal location in air hockey or moving a target block to a goal position in a robotics task. One challenge of goal-conditioned rewards is their sparsity. Without additional inductive biases, making this reward dense is challenging and often requires significant exploration (Andrychowicz et al., 2017; Fang et al., 2019). This sparsity is especially significant in combinatorially complex domains, such as a room with many objects or a scene with numerous state elements. If the desired behavior requires the policy to induce a chain of interactions to achieve the goal, the agent often lacks the opportunity to observe goals and, as a result, does not receive meaningful feedback.

---

*denotes equal authorship.

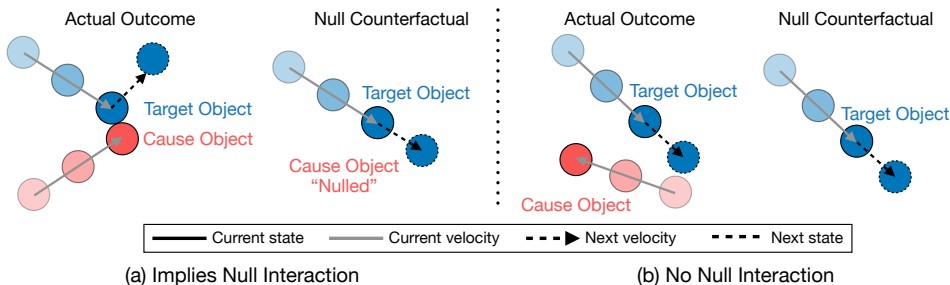

Figure 1: Figure (a) shows a case when a null counterfactual interaction occurs between the cause object and the target object, by comparing the actual event (**left**) with the null counterfactual ("nulled") event (**right**) and observing a *difference* in the target velocity. Figure (b) shows when an interaction does not occur since the actual event *matches* the null counterfactual for the target object.

Hindsight (Andrychowicz et al., 2017) offers a promising direction for receiving rich feedback by relabeling the achieved object state as its intended goal. This allows an agent who may not be achieving many or any of the desired goals to receive learning feedback. Recent methods have added further nuance to hindsight relabeling, allowing for smooth rewards based on distribution matching between the goal and observed state (Ma et al., 2022; Sikchi et al., 2024a). However, while hindsight provides impressive improvements in settings when the goal space is relatively uniform, hindsight still struggles in combinatorial domains. This is because of two fundamental distribution mismatches: (i). the distribution of desired goals often does not match the states reached by the agent early in training; (ii). the distribution of high reward behaviors, i.e. action sequences, under the hindsight distribution often differs significantly from those that induce high rewards under the true test distribution.

To see how this is a challenge for hindsight, consider the example of a robot in a large room. Imagine a robot agent tasked with moving a block to a goal location in a large room, where the starting position of the block and goal are randomly initialized anywhere in that room. Hindsight relabeling will then give high rewards to trajectories where the block does not move at all. However, the desired goals are unlikely to be initialized with the block already inside the goal, so there is a mismatch between the actual and hindsight distribution. Furthermore, when the object is already at the goal, the optimal behavior is to *avoid* interacting with the block so that the agent does not push it out of the goal. This behavior is highly specific to states where the block is already at the goal—in other relative goal positions, it is *necessary* to interact with the block. These intuitions suggest that filling a hindsight buffer with mismatched trajectories will limit sample efficiency.

In this work, we focus on a simple and promising intuition: filtering the hindsight buffer to contain action-dependent interactions. In our example, this will entail only keeping hindsight trajectories where the agent actually pushes the block. These trajectories better match the desired goal distribution, since the agent must have shifted the object from the initial position. They also contain more meaningful behaviors, because by definition the agent must have exerted some control on the target object. However, compared to domain-specific heuristics, such as checking if the target object has moved, interactions can apply to *any* relationship between primitive agent actions, and effects. For example, consider a dynamic domain such as robot air hockey, where the target object (the puck) is moving, but there are often still many trajectories with low-scoring goals (letting the puck fall to the floor) that would have significantly different goal distributions and desired action distributions. By contrast, an interaction-based filtering method requires a principled model of interactions.

Identifying interactions through causal relationships has shown recent headway in several applications (Chuck et al., 2020; 2023; Wang et al., 2023; Hwang et al., 2023), and recently incorporated formalization inspired by actual causality (Chuck et al., 2024b). Under these formulations, an interaction between one object (the cause) and another (the target) is when the cause object induces an effect in the target. In this work, we incorporate a general inductive bias we describe as *null counterfactuals*. Null counterfactuals describe the counterfactual states where an object does not exist (i.e. *nulled*) and everything else is the same. Then, an interaction between the cause and the target occurs when the outcome changes when the cause is nullled. For example, consider Figure 1 where two balls are set to bounce off each other. If removing the red ball changes the outcome, then the red ball is interacting with the blue ball. In this work, we take this intuition, formalize it, and learn models to identify interactions.

To identify the interactions, we introduce the Null Counterfactual Interaction Inference (NCII) Algorithm, which uses a masked dynamics model using trajectories with varying subsets of the causal variables in the environment, and then queries that model to identify the interactions. Then, we introduce the Hindsight Relabeling using Interactions (HInt) algorithm, which uses interaction inference to filter the trajectories added to the hindsight buffer, prioritizing action-induced interactions. Empirically, we demonstrate that NCII matches the performance achieved by prior work in the Random Vectors domain (Hwang et al., 2023; Chuck et al., 2024b), demonstrating the efficacy of the method, and performs well in simulated robotics using Robosuite (Zhu et al., 2020), Robot Air Hockey (Chuck et al., 2024a), and Franka Kitchen (Gupta et al., 2019). Next, we demonstrate that HInt improves RL performance compared to recent baselines of GCRL and Causal RL baselines, even when using interactions it identifies, through evaluations on variations of goal-conditioned Spriteworld, Robot Pushing, Robot Air Hockey, and Franka Kitchen.

To summarize the contributions of this work:

- We introduce the Null Counterfactual Interaction Inference (NCII) for inferring interactions, and the Hindsight Relabeling using Interactions (HInt) algorithm for improving hindsight in GCRL.
- We provide an empirical evaluation of NCII compared to existing actual cause inference methods in Random Vectors, Spriteworld, Robosuite, Robot Air Hockey, and Franka Kitchen domains when using ground truth variable state
- We evaluate the efficiency of HInt applied to GCRL in Spriteworld, Robosuite, Robot Air Hockey, and Franka Kitchen.

## 2 RELATED WORK

Interactions have been applied in a variety of ways for reinforcement learning (Buesing et al., 2018), particularly for understanding compositional relationships between objects and causal relationships among object interaction events. Several analogous ideas have been proposed in the RL and robotics literature for detecting interactions between objects, including local causality (Pitis et al., 2020) and controllability (Seitzer et al., 2021). Other related approaches include identifying causes using changepoint detection (Chuck et al., 2020), Granger-causality based tests (Chuck et al., 2023), point-wise conditional mutual information (Seitzer et al., 2021), model gradients (Wang et al., 2023), graph networks (Feng & Magliacane, 2024), context-specific invariance (Hwang et al., 2023), contacting inference for rigid bodies (Manuelli & Tedrake, 2016; Liu et al., 2024) and using interventional data (Baradel et al., 2019; Lippe et al., 2023). These methods then use interactions in the context of hierarchical reinforcement learning (Chuck et al., 2023), exploration (Seitzer et al., 2021; Wang et al., 2023), model-based RL (Wang et al., 2022) and data augmentation in RL (Pitis et al., 2020; 2022; Urpí et al., 2024). The HInt algorithm is a novel application of interactions to hindsight relabeling in goal-conditioned reinforcement learning, while NCII is a novel counterfactual inference algorithm for identifying factor interactions.

Recent work by Chuck et al. (2024b) proposes a unification of the correlational and heuristic definitions of state-specific cause-effect relationships with those described by actual cause. The actual cause problem identifies the variables that are the cause of an effect in a particular state, which is a causal interpretation of factor interactions. Our work builds upon prior definitions of actual causation (Pearl, 2000; Halpern, 2016), which have often used contrastive necessity (Beckers, 2021) to define actual causes. However, these methods are not typically applied outside the context of discrete states with a small state space. Unlike these methods, NCII includes an assumption about the "null counterfactual state" of objects, which makes the problem more tractable, though it assumes an inductive bias about the existence of a null state.

Goal-conditioned Reinforcement Learning (GCRL) (Puterman, 1990; Kaelbling, 1993) has been investigated as a way to learn multiple behaviors from sparse goal-reaching reward (Liu et al., 2022). GCRL has seen significant success learning to achieve complex behaviors (Chane-Sane et al., 2021) through image-based goals (Nair et al., 2018) or goals in a learned latent space (Khazatsky et al., 2021). The last work is particularly relevant, since it generates interaction goals, though from prior experience. Hindsight experience replay (HER) (Andrychowicz et al., 2017) or generalized HER (Li et al., 2020) reduces the challenge of learning from sparse rewards by relabeling failures with the achieved state. However, this biases the distribution of goals to those visited by the agent (Lanka & Wu, 2018), and induces bias in the visited states in the replay buffer (Bai et al., 2021). Both

theoretical analysis (Zheng et al., 2024) and empirical methods have been applied to modify the hindsight resampling strategy, including curriculum-based sampling (Fang et al., 2019), curiosity-based sampling (Zhao & Tresp, 2019) and maximum-entropy regularized sampling (Zhao et al., 2019). HInt offers an alternative and complementary method by using interactions as an inductive bias on useful states for resampling. By modifying the distribution of hindsight goals to those with interactions, HInt draws parallels with distributional representations of GCRL (Ma et al., 2022; Sikchi et al., 2024b) In this context, modifying hindsight is an alternate way of providing dense signal (Agarwal et al., 2023) while still matching the desired goal distribution.

## 3 PROBLEM FORMULATION

### 3.1 GOAL-CONDITIONED REINFORCEMENT LEARNING

A Markov Decision Process (MDP) is formalized with the tuple $(\mathcal{S}, \mathcal{A}, r, p)$, where $\mathbf{s} \in \mathcal{S}$ is a state in a state space. In this work, we focus on *Factored* MDPs (FMDP) (Kearns & Koller, 1999; Boutilier et al., 2013), where the state space is factored into $n$ factors: $\mathcal{S} \coloneqq \mathcal{S}_1 \times \ldots \times \mathcal{S}_n$. $a \in \mathcal{A}$ is an action in the action space, and $p(\cdot|\mathbf{s}, a)$ is the transition probability. The reward function is $r : \mathcal{S} \times \mathcal{A} \to \mathbb{R}$. In the goal-conditioned RL setting (Kaelbling, 1993), $g \in \mathcal{G}$ is a goal in the space of goals, and $r : \mathcal{S} \times \mathcal{G} \to \mathbb{R}$ is a sparse, goal-conditional reward. The policy $\pi(a|\mathbf{s}, g)$ outputs a distribution over actions conditioned on the state and goal. The objective of RL is to maximize the expected return. For a trajectory $\tau$ defined as a sequence $(\mathbf{s}_0, a_0, \ldots, \mathbf{s}_T, a_T)$, space of trajectories $\mathcal{T}$, distribution of trajectories induced by a policy $\pi$ as $d_{\pi, \mathcal{T}}$ and goal $g$, the objective of goal-conditioned RL is to maximize the expected return (over transition dynamics, policy and goal distributions). The expected return of a state and goal is: $\mathrm{ret}(s, g) \coloneqq E_{g \sim \mathcal{G}, \tau \sim d_{\pi, \mathcal{T}}} \sum_{t=0}^{T} [\gamma^t r(s_{t+1}, g) | s_0 = s]$.

One key insight in practically applying goal-conditioned RL is *hindsight*, where the desired goal is replaced in hindsight with the actual goal reached by the agent. Note that in this setting, we operate under the formulation where the goal space does not need to be the same as the state space. For example, the goal space is the state of a single state factor. Formally, describe $g_\delta \sim d_{\delta, \mathcal{G}}$ as the *desired goal*, sampled from the distribution of desired goals $d_{\delta, \mathcal{G}}$. This distribution is assigned by the goal-conditioned RL setting and assigned at the start of the trajectory. By contrast, hindsight uses goals sampled from the space of states induced by the policy. $P(\mathbf{s}|\pi)$ is the probability of a state given a policy, we can describe this distribution of hindsight goals induced by $\pi$ as $d_{\pi, \mathcal{G}} \coloneqq P(\cdot|\pi)$. Then, Hindsight relabeling induces the distribution of goals $d_{\mathrm{hind}} = \mathrm{Mix}_\alpha(d_{\delta, \mathcal{G}}, d_{\pi, \mathcal{G}})$, where $\mathrm{Mix}_\alpha$ denotes a mixture distribution with coefficient $\alpha$: $\mathrm{Mix}_\alpha(\mu_1, \mu_2) = \alpha \mu_1 + (1 - \alpha) \mu_2$ [1].

### 3.2 INTERACTION INFERENCE

In order to identify interactions, this work builds on formulations from Chuck et al. (2024b) to describe an interaction as a directed connection between a set of state factors $\mathbf{X}$ drawn from $S_1, \ldots S_n$ and a particular outcome $S'_j = \mathbf{s}'_j$. In previous work, identifying this edge requires identifying a globally minimal invariant set of the cause state, which is intractable. This work replaces that with the following *null assumption*:

**Definition 3.1** (Null Counterfactual Interaction Assumption). Suppose that for any state factor $S_i$, there exists a null state $\mathbf{s}_{i,\circ}$, which is when that factor is not present. Define $\mathbf{s} \circ \mathbf{s}_i$ as the counterfactual state where $\mathbf{s}$ is exactly the same except for $S_i = \mathbf{s}_{i,\circ}$. Then if the observed transition has a different probability under this state:

$$p(S_j = \mathbf{s}'_j | S = \mathbf{s}) \neq p(S_j = \mathbf{s}'_j | S = \mathbf{s} \circ \mathbf{s}_i), \tag{1}$$

$S_i$ is considered one of the state factors interacting with $S_j$ in state transition $\mathbf{s}, \mathbf{a}, \mathbf{s}'$. We describe this comparison as $S_i$ being "nulled."

This definition describes the intuition in Figure 1, where a null-counterfactual interaction is defined when the transition is modified by replacing the state of a "cause" object with its null state. Leveraging this definition, NCII takes advantage of the null assumption to learn a model that can infer which state factors are interacting through a counterfactual test.

---

[1] We add the probability mass/density functions, scaling them appropriately to ensure they still sum/integrate to 1.

## 4 METHODS

This section is separated first into the NCII algorithm, which infers interactions by null counterfactual interaction assumption, followed by the HInt method, which uses inferred interactions for hindsight.

### 4.1 NULL COUNTERFACTUAL INTERACTION INFERENCE ALGORITHM

The objective of the NCII algorithm is to identify which state factors are interacting in a given transition. Formally, take a factored state transition $\mathbf{s}, \mathbf{a}, \mathbf{s}'$ and identify an $n \times (n+1)$ matrix $\mathbb{B} \in [0,1]^n \times [0,1]^{n+1} := \mathcal{B}$, where $\mathbb{B}_{ji} = 1$ indicates that $S_i$ is interacting with $S_j$ in the state transition, and $\mathbb{B}_j$ is the row of all state factors interacting with $S_j$. The $n+1$ column corresponds to the actions.

Null-counterfactual interactions are defined by the difference between the next null-counterfactual state $\hat{s}'_j$ and the actual next state $s'_j$. Since we cannot actually observe the counterfactual outcome, we instead query a learned forward dynamics model. NCII learns a masked forward model $f : \mathcal{S} \times \mathcal{A} \times \mathcal{B} \to P(\cdot|S, A)$, which predicts the distribution over the next state. The prediction of $\mathbf{s}_j$ is masked by $\mathbb{B}_{\cdot j}$, so if $\mathbb{B}_{ij} = 0$, this is equivalent to factor $S_i$ not being present in that state—taking on the null state. We can then query this model to identify whether an interaction occurs in a state if nulling a variable significantly changes the probability of the observed outcome $\mathbf{s}$.

To train the masked forward model $f(\mathbf{s}, \mathbf{a}, \mathbb{B}; \theta)$, we assume that the null state exists in the dataset. We include this in two ways: first, we assume that in each trajectory $\tau_k$, there is a subset of factors $\mathbf{V}^k$, represented with a binary vector. Then, each state tuple is augmented with $\mathbf{v}$ to give $(\mathbf{s}, \mathbf{v}, \mathbf{a}, \mathbf{s}')$. Obviously, not all tasks exhibit this property, and we discuss an alternative method that relies on passive modeling to simulate this null effect in Appendix D. Given a dataset $\mathcal{D}$ of state-valid-action-next state tuples, we can construct a valid-dependent matrix $\mathbb{B}(\mathbf{v})$ where the column $i$ is zero if $\mathbf{v}_i = 0$. This corresponds to a trajectory where the object $i$ is not present, which is only possible in settings where each trajectory can contain a different subset of the state factors. Then we learn $f$ by maximizing the log-likelihood of the outcome of the dataset, where $f(\mathbf{s}, \mathbf{a}, \mathbb{B}(\mathbf{v}); \theta)[\mathbf{s}']$ denotes the probability of $\mathbf{s}'$:

$$\max_\theta \sum_{(\mathbf{s}, \mathbf{v}, \mathbf{a}, \mathbf{s}') \sim \mathcal{D}} \log f(\mathbf{s}, \mathbf{a}, \mathbb{B}(\mathbf{v}); \theta)[\mathbf{s}']. \tag{2}$$

The null operation for state factor $S_j$ is then a comparison between the forward model with and without nulling of the $i$th causal variable. This is performed by defining $\mathbb{B}(\mathbf{v})_j \circ S_i$ as the row with index $\mathbb{B}(\mathbf{v})_{ji} = 0$. Then we can compare the predicted log-likelihoods of the actual state $\mathbf{s}_j$, and identify an interaction if the difference is greater than $\epsilon_{\text{null}}$

$$\text{Null}(\mathbf{s}, \mathbf{a}, \theta)_{ji} = \mathbb{1}\left( \log \left( f(\mathbf{s}, \mathbf{a}, \mathbb{B}(\mathbf{v}); \theta)_j[\mathbf{s}_j] - f(\mathbf{s}, \mathbf{a}, \mathbb{B}(\mathbf{v}) \circ S_i; \theta)_j[\mathbf{s}_j]) \right) > \epsilon_{\text{null}} \right). \tag{3}$$

However, relying only on $\mathbf{v}$ to observe null states can result in a limited distribution. This is because, even if the null state is equivalent, the out-of-distribution nature of the prediction can result in a low predicted log-likelihood, falsely suggesting an interaction when none exists. To combat this, we utilize an iterative joint optimization process where the forward model is retrained with $\mathbb{B}(\mathbf{v})$ replaced by the null counterfactuals, as estimated by an inference function $h : \mathcal{S} \times \mathcal{A} \to \mathcal{B}$.

To learn $h(\mathbf{s}, \mathbf{a}; \phi) = \hat{\mathbb{B}}$, which we call the *interaction model*, we take $\text{Null}(\mathbf{s}, \mathbf{a}, \theta)_{ji}$ at every state under the current forward model as the targets for $h$, and optimize $h$ using binary cross entropy:

$$\min_\phi \sum_{(\mathbf{s}, \mathbf{a}, \mathbf{v}, \mathbf{s}') \sim \mathcal{D}} \text{Null}(\mathbf{s}, \mathbf{a}, \theta) \cdot \log(h(\mathbf{s}; \phi)) + (\mathbf{1} - \text{Null}(\mathbf{s}, \mathbf{a}, \theta)) \cdot \log(\mathbf{1} - h(\mathbf{s}; \phi)). \tag{4}$$

We use the outputs of $h$ instead of $\text{Null}(\mathbf{s}, \mathbf{a}, \theta)_{ji}$ directly because 1) a trained model may extrapolate better as opposed to a statistical test utilizing a possibly inaccurate forward model; 2) inference using the null test is $O(n^2)$ to evaluate each $\mathbb{B}_{ji}$ using the forward model, but $O(1)$ as an inference model; and 3) $h$ can output soft predictions rather than binary values, which make the optimization more smooth. To jointly optimize the forward and null models we iterate between the following two steps:

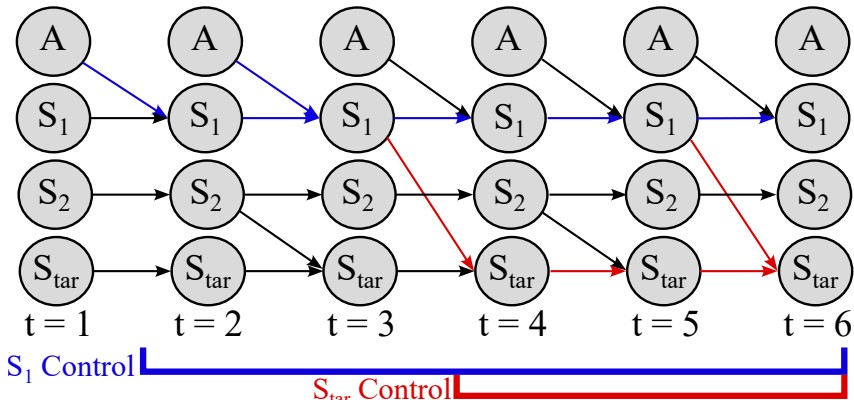

Figure 2: An example of the unrolled dynamic interaction graph, where an edge indicates an interaction $\mathbb{B}_{ji}^{(t)} = 1$ from $t$ to $t+1$. HInt identifies trajectories where the agent exerts control on the target object, as measured by a path in the unrolled interaction graph. The colors indicate the timesteps during which each state factor is controlled: $S_1$ is controlled from $t=2$ to $t=6$, $S_2$ is not controlled, and $S_{\text{tar}}$ is controlled from $t=4$ to $t=6$.

1. Train $f$ with a fixed $h$ to maximize the log probability of the dataset using Equation 2.

2. Train $h$ using Equation 4 to predict the null operation outputs of $f$ computed using Equation 3.

In practice, learning to identify interactions is challenging when interactions are rare events, such as a robotic manipulation domain where the robot rarely touches the object it should manipulate. These challenges have been approached in prior work Chuck et al. (2023; 2024b) by reweighting and augmenting the dataset using statistical error. We include a description of how those methods are adapted and used in this algorithm in Appendix D.

## 4.2 INTERACTIONS FOR HINDSIGHT

Next, we use interactions to select hindsight trajectories (Andrychowicz et al., 2017) for GCRL to improve sample efficiency and performance. As intuition, consider Robot Air Hockey example where the objective is to hit the puck to a goal position, where the environment will reset when the puck hits the agent's side of the table. Vanilla hindsight relabeling will reassign hindsight goals to wherever the puck hits the agent end of the table, which would incentivize missing the puck. This scenario and the block-pushing scenario in Section 1, illustrate situations where hindsight relabeling can hurt performance. Keeping only relabeled trajectories where the agent exerted control on the target object can mitigate this issue. HInt identifies these trajectories using interactions.

Formally, assume a goal-conditioned setting where the goal space consists of a single target factor $i$,[2] such that $g_{\text{ach}} : \mathcal{S} \to \mathcal{S}_i$. Then for any particular trajectory $\tau$ of length $T$ and desired goal $\mathbf{g}_{\text{des}}$, the hindsight (relabeled) goal is $\mathbf{g}_{\text{rel}} = g_{\text{ach}}(\mathbf{s}^{(t_r)})$, where vanilla hindsight selects $t_r = T$. $\bar{\mathbb{B}} := \{\mathbb{B}^{(1)}, \ldots, \mathbb{B}^{(T)}\}$ is the sequence of interaction graphs for the trajectory. HInt adds the trajectory if it passes a filtering function $\chi(\bar{\mathbb{B}}) \to \iota, t_r$, where $\iota$ is a binary decision to reject $\tau$ from the hindsight buffer, and $t_r \in \{1, \ldots, T\}$ is the timestep of a suitable hindsight goal.

Intuitively, HInt should keep trajectories where the agent's behavior is *responsible* for the outcome. Interactions give us a factored way of identifying this responsibility. Each edge $(i, j)$ in the interaction graph $\mathbb{B}^{(t)} \in \mathcal{B}$ denotes that factor $i$ exerts an effect on factor $j$. In the time unrolled graph, where $\mathbb{B}^{(t)}$ are the edges at time $t$ (see Figure 2), a path in the unrolled graph between actions and the target object implies a sequence of control where the agent exerted an effect on the target object.

To identify this path, construct the temporal graph corresponding to the trajectory, where there are $n+1$ nodes for each timestep, and $\mathbb{B}^{(t)}$ describes the edges from timestep $t$ to timestep $t+1$. Note that the "passive" edge represented by $\mathbb{B}_{jj}$ is always 1. Then, we identify the set of timesteps $\mathcal{T}_{\text{inter}}$ as

---

[2]without loss of generality to multiple factor goals

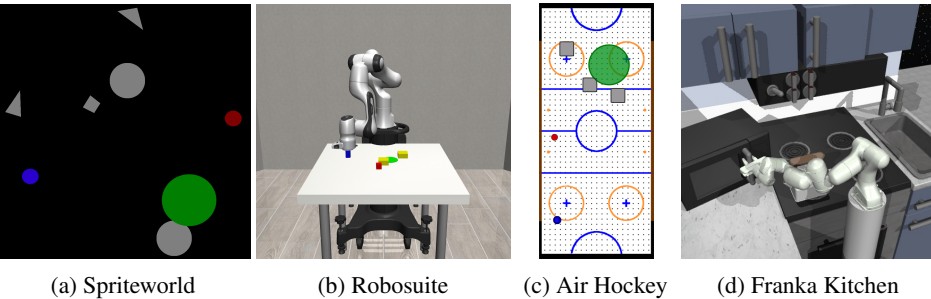

|  (a) Spriteworld | (b) Robosuite | (c) Air Hockey | (d) Franka Kitchen |

Figure 3: Visualizations of domains used for evaluation. Goals are in green and target objects are in red (except for Franka Kitchen domain).

those for which there is a path from the action node $n + 1$ to the target variable $S_i$. The states are visualized in an example in Figure 2. Then, $t_r$ is sampled from $\mathcal{T}_{\text{inter}}$. If this set is empty, then $\iota = 0$, i.e. the trajectory $\tau$ is rejected for hindsight relabeling. In practice, identifying this path can suffer from compounding model errors, since it relies on accurate interaction inference for every factor, so we propose several simpler alternatives in Appendix E. In particular, we limit the length $l = 3$ of a chain in the graph including actions, which we call the "control-target interaction" criteria.

Putting it all together, the Hindsight Relabeling using Interactions algorithm is summarized in Algorithm 1 and Appendix Figure 6.

---

**Algorithm 1 Hindsight relabeling using Interactions (HInt)**

---

**Input:** Goal-conditioned MDP $E$ with target variable $S_i$. Learned Null Model $f(\mathbf{s}, \mathbf{a}, \mathbb{B}; \theta), h(\mathbf{s}; \phi)$
**Initialize** Dataset $\mathcal{D}$, hindsight dataset $\mathcal{H}$ as empty sets of state transitions. Off-policy Goal-conditioned RL algorithm $\mathbb{A} : \psi \times \mathcal{D} \to \psi$ which outputs policy $\pi(\mathbf{s}, \mathbf{a}, \mathbf{g}; \psi)$.
**repeat**
  **Policy Rollout**: Utilize Goal-conditioned RL policy $\pi(\mathbf{s}, \mathbf{a}, \mathbf{g}_{\text{des}}; \psi)$ and null model $f$ to collect trajectory $\tau$ and interactions $\bar{\mathbb{B}} := \{\mathbb{B}_1, ... \mathbb{B}_T\}$ and add $\tau$ to $\mathcal{D}$.
  **Hindsight Relabeling**: Test $\chi(\bar{\mathbb{B}}) \to \iota, t_r$ and update $\mathcal{H}$ with the trajectory $\tau$ if $\iota = 1$, selecting hindsight goal $g_{\text{ach}}(\mathbf{s}^{(t_r)})$ sampled uniformly from $\mathcal{T}_{\text{inter}}$.
  **Policy Update**: update policy $\psi \leftarrow \mathbb{A}(\psi, \mathcal{D} \cup \mathcal{H})$
**until** $\psi$ converges

---

## 5 EXPERIMENTS

In this section, we aim to answer the following questions: **(1)** How does the NCII perform compared with existing interaction inference algorithms? **(2)** Does Goal-conditioned RL benefit from hindsight relabeling using interactions? **(3)** How does the performance of HInt compare when using inferred interactions from NCII vs ground truth interactions? The first question is evaluated in Section 5.1, and the other two in Section 5.2.

Before going to the results, we provide some details on the domains visualized in Figure 3. Each domain has several variations that change the number of factors (an "obstacles" variant) and the size of the factors (a "small" variant) in physical domains (making interactions less frequent). These changes are further discussed in Appendix H.

**Random DAG Null** (Hwang et al., 2023; Chuck et al., 2024b) is a simple domain to test interactions represented by arbitrary functional relationships between state factors. For the parents of $S_j$ represented with $\mathbf{X}$, normally distributed noise variable $\nu_j \sim \mathcal{N}(0, 1)$ and $[\mathbf{s}_i, \mathbf{s}_j]$ denotes the concatenation of the two vectors, the next state $S'_j$ is determined by:

$$\mathbf{s}'_j := \begin{cases} \frac{1}{|\mathbf{X}|} \sum_{\mathbf{s}_i \in \mathbf{x}} \mathbb{1}(c_i^\top [\mathbf{s}_i, \mathbf{s}_j] > 0) \cdot A_i [\mathbf{s}_i, \mathbf{s}_j] & \exists s_i \quad \text{s.t.} \quad c_i^\top [\mathbf{s}_i, \mathbf{s}_j] > 0 \\ A_j \mathbf{s}_j & \text{otherwise} \end{cases}. \tag{5}$$

To support nulling, each length 50 trajectory randomly samples a subset of the factors. This domain assesses whether inference algorithms can recover arbitrary linear interaction relationships in the null setting. $n-$in indicates a random DAG domain with $n$ factors.

**Spriteworld Null** (sprite) (Watters et al., 2019) is a suite of environments based on Spriteworld, where 2D polygons and circles collide in a frictionless 0-gravity environment. To make this domain goal-conditioned, a "target" sprite is used to define the goal space, and the agent controls a single sprite to manipulate the target to the goal. This domain assesses challenging control of striking and securing the frictionless target in the goal. Sprite-$n$ indicates a Spriteworld with $n$ objects.

**Robosuite** (robo) (Zhu et al., 2020) is a 3D robotics domain where the agent controls a 9-DOF robot arm through end effector control to move a block to a target location in the midst of obstacles. This domain assesses a 3D tabletop pushing domain with quasistatic dynamics (the objects do not move unless manipulated).

**Air Hockey** (air) (Chuck et al., 2024a) is a dynamic robotics domain where the agent controls a paddle to strike a puck to a desired target location with constant gravity pulling the puck towards the bottom of the table. The domain resets if the puck hits the bottom of the screen. This domain assesses a 2D dynamic domain where some achieved goals (the puck at the bottom) are never desired goals (all goals are initialized on the upper half of the table).

**Franka Kitchen** (kitchen) (Gupta et al., 2019) is a 9-DoF Franka robot arm to complete a series of tasks in a simulated kitchen environment, including opening the microwave, turning on the light switch, and opening the sliding cabinet. This domain asses the control over a 3D simulated kitchen where the robot needs to interact with various objects to achieve the desired goal configuration.

In Spriteworld, Robosuite, Air Hockey, and Franka Kitchen, instead of using explicit null trajectories we use simulated nulling, as described in Appendix D.

## 5.1 INFERENCE EXPERIMENTS

When assessing inference, we compare against several previous methods that cover a wide range of techniques for identifying interactions: JACI (Chuck et al., 2024b), gradients (Wang et al., 2023), attention weights (Pitis et al., 2020), and Neural Contextual Decomposition (NCD) (Hwang et al., 2023). We test NCII with two architectures, Pointnet (Qi et al., 2017) and Graph Neural Network (GNN) (Scarselli et al., 2008; Kipf & Welling, 2017) based architecture, which we include additional details for in Appendix G.

**Null Inference**: To answer question **(1)**, if the null assumption achieves statistically significant reduction in misprediction rate when compared with the baselines, we perform a comparison on the above domains. We collect a fixed dataset of 1M states for random DAG and 2M states for Spriteworld, Robosuite Air Hockey, and Franka Kitchen. Then we compare accuracy at recovering the ground truth interactions, described by contacts in the physical domains, and $\mathbb{1}(c_i^\top[\mathbf{s}_i, \mathbf{s}_j] > 0)$ in Random DAG Null. Each method is trained over 5 seeds until convergence. As we observe in Table 1 provided with null data or simulated nulling, this inference method achieves statistically significant reduction in misprediction rate compared to existing baselines.

## 5.2 HINDSIGHT RELABELING USING INTERACTIONS

Before describing the empirical evaluation of GCRL, we briefly describe the RL baselines we compare against for this work:

**Vanilla**: Basic offline goal conditioned RL using Deep Deterministic Policy Gradient (DDPG) (Lillicrap, 2015) without hindsight.

**Hindsight** (Andrychowicz et al., 2017): Vanilla hindsight gives a baseline for adding hindsight in the context of rare-interaction domains.

**Prioritized Replay** (Schaul et al., 2015): Prioritized replay with hindsight uses TD error to prioritize high error states, assessing if policy error is sufficient to emphasize desirable states.

**f-policy gradients (f-pg)** (Agarwal et al., 2023): f-divergence policy gradients use a distributional perspective on goal reaching. From that perspective, HInt can be seen as modifying the hindsight

| Method | NCII w/ Point | NCII w/Graph | JACI | Gradient | Attention | NCD |
|---|---|---|---|---|---|---|
| 1-in | **0.8 ± 0.2** | **1.0 ± 0.1** | 1.6 ± 0.1 | 4.5 ± 2.5 | 4.8 ± 1.6 | 30.3 ± 2.9 |
| 2-in | **1.4 ± 0.2** | **1.4 ± 0.2** | **2.0 ± 0.2** | 33.9 ± 1.0 | 27.3 ± 5.2 | 22.1 ± 3.4 |
| 6-in | **25.1 ± 5.8** | **26.3 ± 5.2** | **27.3 ± 4.7** | 40.2 ± 0.8 | 35.4 ± 3.9 | 31.1 ± 0.5 |
| Sprite-2 | **4.5 ± 1.2** | **6.2 ± 3.9** | 15.3 ± 2.1 | 13.7 ± 0.9 | 28.6 ± 4.0 | 25.3 ± 1.7 |
| Sprite-2 rare | **6.5 ± 4.5** | **6.9 ± 3.4** | 15.5 ± 2.6 | 14 ± 0.4 | 30.2 ± 5.1 | 23.2 ± 2.5 |
| Sprite-6 | **7.4 ± 4.3** | **8.3 ± 2.9** | 30.6 ± 1.3 | 24 ± 0.3 | 18.4 ± 3.3 | 16.0 ± 1.2 |
| Air Hockey | **11.3 ± 1.3** | **13.9 ± 2.8** | 16.9 ± 2.3 | 47.5 ± 0.6 | 39.8 ± 3.5 | **14.3 ± 1.4** |
| Robosuite | **5.1 ± 0.3** | **7.2 ± 1.6** | **6.2 ± 0.5** | 40.9 ± 1.2 | 43.7 ± 5.6 | 49.5 ± 3.3 |
| Kitchen | **7.2 ± 1.4** | **10.0 ± 3.2** | 19.7 ± 2.2 | 34.6 ± 1.3 | 21.8 ± 2.0 | 39.2 ± 6.4 |

Table 1: Misprediction rate (lower is better) with standard error of inference in evaluated domains from state. Interactions are reweighted to be 50% of the test dataset. Boldface indicates within $\sim 1$ combined standard error of the best result.

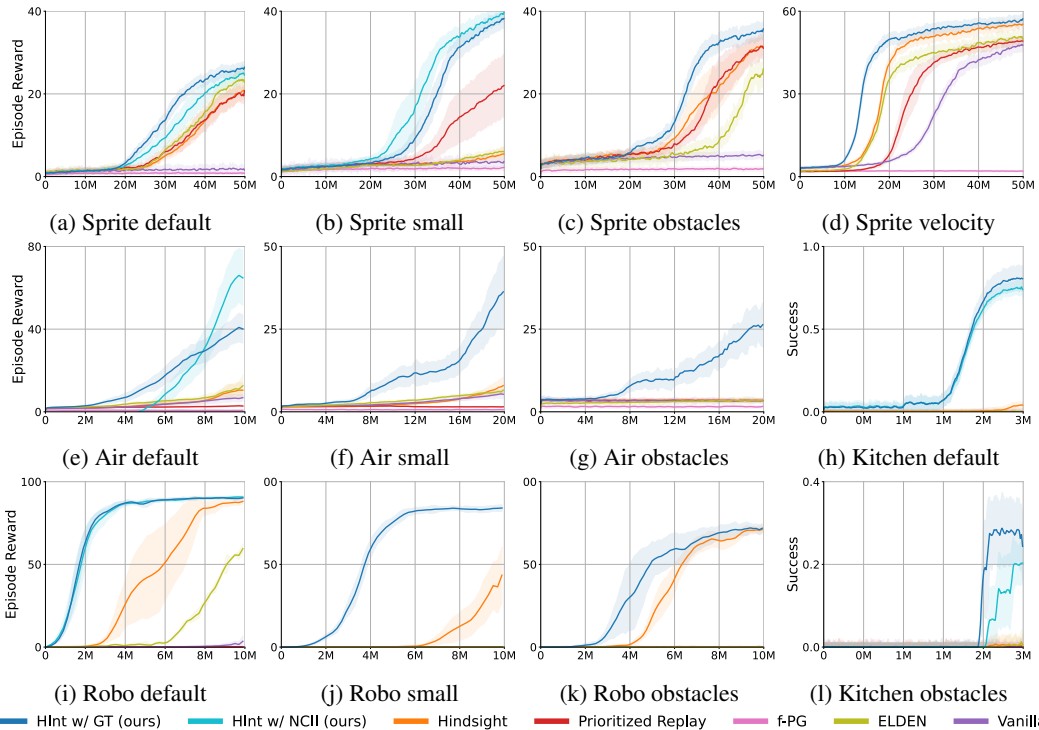

Figure 4: Comparison of HInt and HInt with NCII against baselines, 5 trials for each. HInt with NCII is used in Spriteworld default, small, Robosuite default, Air Hockey default, Kitchen default, and obstacles. The shading indicates standard error. X axis is the number of timesteps.

distribution to better match the distribution of desired goals since desired goals will tend to be gated by an interaction. f-pg explicitly directs behavior towards the desired goal distribution, but without using interactions.

**Exploration via Local Dependencies (ELDEN)** (Wang et al., 2023): This method uses gradient-defined caused-based interactions for exploration with the ensemble of learned forward models. This compares interactions with hindsight against interactions for exploration.

**Causal Action Influence (CAI)** (Seitzer et al., 2021): This method uses pointwise mutual information with actions to incentivize exploration of controllable states. This compares chains of factor interactions with pure action controllability. Results of CAI are shown in Appendix Figure 12.

### 5.2.1 HINDSIGHT EVALUATION

Now, we can provide empirical evidence towards **(2)** Does GCRL benefit from filtering resampling with actual cause graphs? We evaluate Goal-conditioned RL in 3 of the modified Spriteworld domains,

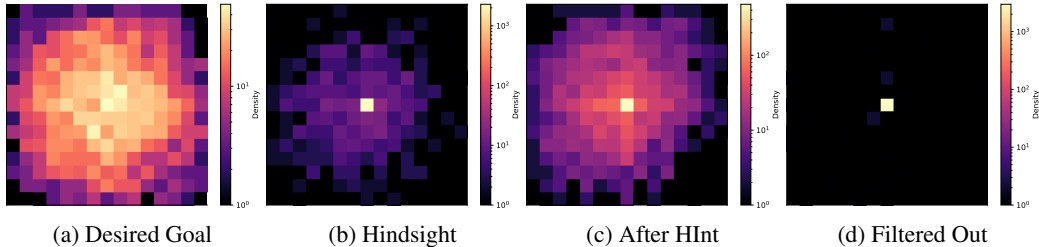

| (a) Desired Goal | (b) Hindsight | (c) After HInt | (d) Filtered Out |

Figure 5: Relative position heatmap between initial state and **a)** sampled or "desired" goal, **b)** hindsight goals, **c)** goals after HInt filtering, **d)** goals removed by HInt, over 3000 goals in Spriteworld default.

3 modified air hockey domains, and 3 modified Robosuite domains and Franka Kitchen. This variety of domains allows us to assess a wide range of interaction rates, kinds of dynamics (collision, quasistatic, articulated, etc.), and numbers of state factors. In this work we applied HInt using the action graph filtering strategy in all the domains except the obstacles variants, where we found the control-target graph filtering strategy was more stable. Training details and hyperparameters can be found in Appendix I. As we see through the training curves in Figure 4, HInt with ground truth interactions or learned interactions achieves as-good-as or better sample efficiency than existing algorithms, demonstrating an up to $4\times$ performance improvement in sample efficiency. While inference is not perfectly accurate at detecting contact, we hypothesize that filtering can occasionally improve accuracy by rejecting contact that only produces minute changes in the target object as a result of control. We also visualize in Figure 5 the proposed source of this performance benefit: filtering out trivial goals, such as those where the target object is initialized (in hindsight) at the goal. This clearly visualizes how this distribution is better matched after HInt filtering compared to the hindsight distribution. Lastly, we also visualize some learned policy rollouts in Appendix M.

### 5.2.2 Hindsight Filtering using NCII

Finally, we can provide empirical evidence for **(3)** How does the performance of HInt compare when using inferred interactions from NCII vs ground truth interactions? We use a trained NCII model in four domains: Spriteworld default, Spriteworld small, Robosuite default, Air hockey default and Franka Kitchen Default. In Figure 4, we see that interactions identified by NCII are equivalent or sometimes exceed ground truth interactions as indicated by contact between objects. We also plot just HInt with ground truth interactions and HInt with NCII interactions in Appendix Figure 7.

## 6 Conclusion, Limitations, and Future Work

Many real-world tasks, from object pushing to dynamic striking, involve sparse interactions in large spaces. Learning a goal-conditioned policy to achieve arbitrary goals in these tasks is often challenging, especially without injecting specific domain information such as distance, and intelligent design of the reward function. Furthermore, because hindsight can fall prey to adding trivial rewards, it can also struggle in these domains. This work demonstrates the effectiveness of applying interactions to improve sample efficiency in GCRL through the HInt algorithm. As interactions become more rare, the additional information provided by HInt through interactions becomes more useful. HInt relies on fundamentally different assumptions from those used in other GCRL methods, future work can investigate integrating it with existing techniques. The effectiveness of HInt is contingent on how the interactions are inferred, and this work also demonstrates a powerful physical inductive bias for interaction identification: the null assumption. Leveraging this assumption, NCII uses learned counterfactual models to improve interaction inference. While this leaves some open questions, NCII sufficiently improves the accuracy of inference compared with existing methods, so that it can be integrated with HInt. Altogether, this work offers a promising intuitive tool for improving the performance of GCRL in real-world environments.

**Limitations.** While HInt is designed to model interactions effectively, it may have limited utility in domains where interactions are less critical, such as certain locomotion tasks. In some cases, interactions might even be detrimental, as in driving or drone navigation. However, in these scenarios, a model that effectively captures interactions can still be highly beneficial, as it helps to understand and avoid potential collisions or conflicts.

ACKNOWLEDGEMENTS

This work has taken place in part in the Safe, Correct, and Aligned Learning and Robotics Lab (SCALAR) at The University of Massachusetts Amherst. SCALAR research is supported in part by the NSF (IIS-2323384), the Center for AI Safety (CAIS), and the Long-Term Future Fund. This work also took place in the Machine Intelligence through Decisionmaking and Interaction Lab at The University of Texas at Austin, supported in part by NSF 2340651, NSF 2402650, DARPA HR00112490431, and ARO W911NF-24-1-0193. The work was supported by the National Defense Science & Engineering Graduate (NDSEG) Fellowship sponsored by the Air Force Office of Science and Research (AFOSR).

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

# APPENDIX
# NULL COUNTERFACTUAL FACTOR INTERACTIONS FOR GOAL-CONDITIONED REINFORCEMENT LEARNING

## TABLE OF CONTENTS

## A    REPRODUCIBILITY STATEMENT

We provide the implementation of NCII, HInt, and other baselines in the appendix code. The installation instructions, detailed settings, and configurations for the data generation process for the benchmarks and datasets can be found in Appendix H and the appendix code. The training details, including hyperparameter settings, experimental setups, and platforms, are provided in Appendix I.

## B    FLOW DIAGRAM

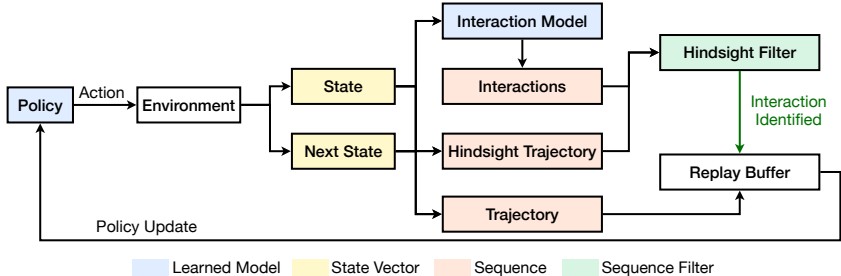

Figure 6: The data flow for the HInt method. An interaction matrix $\mathbb{B}$ is generated at each time step, and at the end of each trajectory is used to filter non-interacting trajectories.

## C    VISUALIZATION OF HINT USING GROUND TRUTH OR NCII INTERACTIONS

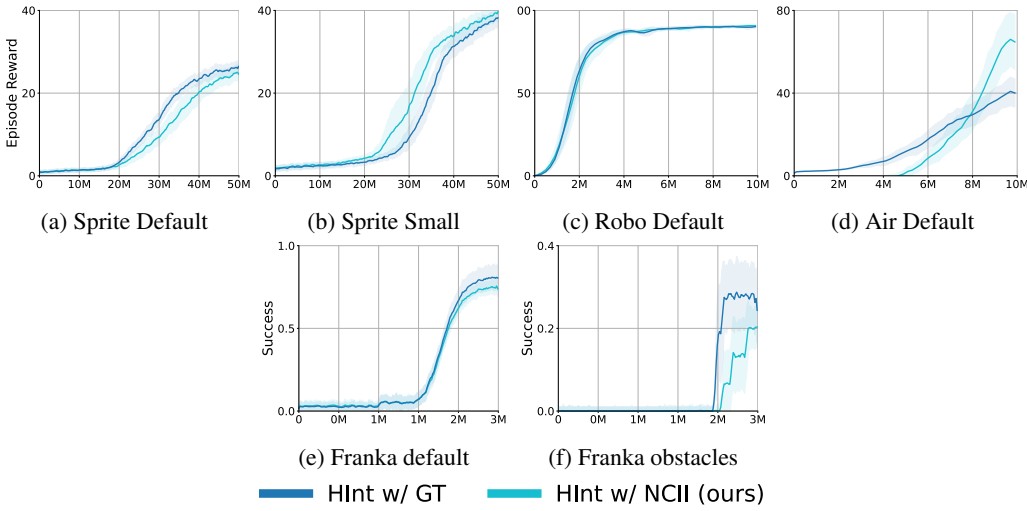

Figure 7: Comparison of HInt and HInt with NCII 5 trials for each. The shading indicates standard error.

## D    SIMULATED NULLING

Nulling is a powerful tool for lifelong learning settings, where the agent will experience a large number of trajectories in a wide variety of settings. However, many RL domains often only contain a few, specific factors and have a sparse number of interactions. It can be expensive to observe every distribution of possible states in this setting. However, we can often leverage the fact that interactions are rare to identify high-priority states using a passive error, a technique introduced by Chuck et al. (2023).

Intuitively, we can estimate states where there is unlikely to be an interaction affecting the target object by observing if the target object exhibits unexpected behavior. If it *does not*, then there is likely *not* an interaction. For example in an object-based domain, an object is probably *not* interacting

with any other object if it maintains its velocity. More specifically, if an autoregressive model $f : \mathcal{S}_j \rightarrow \mathbb{P}(S_j'|S_j)$ is a good predictor of the next target object state $\mathbf{s}_j'$ using only the current target object state $\mathbf{s}_j$, then there is most likely not an interaction. The passive signal is then just the log-likelihood of the observed outcome under the distribution induced by the forward model:

$$e_j^{\text{passive}}(\mathbf{s}_j, \mathbf{s}_j') := \log f(\mathbf{s}_j)[\mathbf{s}_j'] \tag{6}$$

Where $f(\mathbf{s}_j)[\mathbf{s}_j']$ is the probability of $\mathbf{s}_j'$ under the distribution from $f(\mathbf{s}_j)$. When the passive signal is high, this suggests a non-interaction.

We use the passive signal in two ways, first, as a substitute for the null signal. This is because, assuming that there is not any interaction with the target factor in state $\mathbf{s}$, we can null out any other state under the null assumption (if there is no interaction, then the outcome distribution should equal the null distribution). The passive signal can then be used as a proxy for this circumstance since low passive error implies non-interaction. In practice, this means taking the null vector $\mathbf{v}$, which again is a $0 - 1$ vector indicating which objects are present, and randomly zeroing certain indices with some probability. This gives the passive-adjusted null vector $\hat{\mathbf{v}}$:

$$\hat{\mathbf{v}} := \begin{cases} \mathbf{v} \cdot \text{Bernolli}(1 - \epsilon_{\text{sim-null}}) & e_j^{\text{passive}}(\mathbf{s}_j, \mathbf{s}_j') > \epsilon_{\text{passive}} \\ \mathbf{v} & \text{otherwise} \end{cases} \tag{7}$$

Where $\epsilon_{\text{sim-null}}$ is the probability of nulling out a value in high passive likelihood states, and $\epsilon_{\text{passive}}$ is the log likelihood threshold to be considered high likelihood. We can then train using NII using $\hat{\mathbf{v}}$ instead of $\mathbf{v}$.

The second way we use the passive error is to modify the frequency of interaction states observed during training. In many of the domains, interactions are very infrequent, ranging from $0.5\%$ to $0.01\%$. A model learned with these is likely to simply ignore interactions entirely. Because interactions are low-likelihood, however, they also tend to have low likelihood under the passive model $f$. Consequently, we upweight low passive likelihood states so that they are sampled on average $20\%$ of the time. This is a technique used in both Chuck et al. (2023; 2024b).

## E  FILTER CRITERIA

The action path criteria for $\chi$ defined in section 4.2 will capture any control exerted by the agent but has three weaknesses. First, because it incorporates information about all edges, it can be susceptible to false positives, where a false positive edge in a different object could induce a path. As an example, imagine the Spriteworld domain with obstacles. If there is a false positive between an object that was controlled and an obstacle that happened to collide with the target object, this would result in a trajectory misclassified as passing the criteria. Second, again because identifying a path requires all edges, this means using the null counterfactual to generate interactions requires an $O(T \cdot n^2)$ operation. Learning these models accurately can be expensive. Finally, while using a binary signal indicates if *any* action influence was exerted, it does not give a good measure of the *amount* of influence. In practice, this is a challenging problem, but we provide a few heuristic strategies in this section.

First, we define 3 simplifying filtering strategies for $\chi$.

1. **Non-Passive**: This is a permissive strategy that rejects any trajectory that does not have a non-trivial interaction, and assigns a random state after the first non-trivial interaction as the hindsight goal. Formally, construct $\mathbb{B}^{(t)}$ using some subset of the interaction graph and assigning the remaining edges to $0$. Ignore the edges between $(i, i)$, and $(\text{action}, i)$ (because action edges are typically dense). If there is not a nonzero edge remaining, reject the trajectory. $\iota = \sum_{B^{(t)} \in \bar{\mathbb{B}}} \sum_{i=0}^{n} \sum_{j=0}^{n} \mathbb{B}_{ji}^{(t)} = 0$.
   This formulation allows us to utilize a subgraph while still identifying interactions. However, since not all non-passive graphs indicate control, it can be overly permissive.

2. **Non-passive target**: This strategy is a simplification of the general non-passive graph by taking only the row $j$ corresponding to the target object to test for interactions. Formally, for target index $j$, $\iota := \sum_{\mathbb{B}^{(t)} \in \bar{\mathbb{B}}} \sum_{i=1}^{n} \mathbb{B}_{ji}^{(t)} = 0$.

| Domain | $\epsilon_{\text{null}} = 0.1$ | $\epsilon_{\text{null}} = 0.3$ | $\epsilon_{\text{null}} = 0.5$ | $\epsilon_{\text{null}} = 0.7$ | $\epsilon_{\text{null}} = 1.0$ | $\epsilon_{\text{null}} = 1.5$ | $\epsilon_{\text{null}} = 2.0$ | $\epsilon_{\text{null}} = 2.5$ | $\epsilon_{\text{null}} = 3.0$ | $\epsilon_{\text{null}} = 3.5$ |
|---|---|---|---|---|---|---|---|---|---|---|
| 1-in | $4.4 \pm 3.0$ | $3.4 \pm 1.8$ | $3.2 \pm 0.5$ | $1.1 \pm 1.3$ | $1.2 \pm 3.1$ | $1.6 \pm 1.3$ | $1.8 \pm 0.4$ | $2.2 \pm 0.8$ | $2.4 \pm 0.9$ | $2.6 \pm 2.5$ |
| Sprite-1 | $5.2 \pm 2.5$ | $4.9 \pm 0.4$ | $2.8 \pm 2.5$ | $4.6 \pm 3.3$ | $6.4 \pm 5.0$ | $21.6 \pm 12.5$ | $19.6 \pm 4.1$ | $27.4 \pm 3.1$ | $39.5 \pm 4.7$ | $43.5 \pm 17.9$ |

Table 2: Ablation on $\epsilon_{\text{null}}$ parameter (horizontal) for NCII. Note that because $\epsilon_{\text{null}}$ indicates a difference in normalized log-likelihood, the parameter is quite robust. Each is run over 3 seeds.

This has the advantage of reducing the computation to a single row of the graph, though it is still overly permissive. However, since a non-passive interaction *must* occur for a control interaction to occur, it will not filter out any interacting trajectories.

3. **Control-target interaction**: This strategy is the most stringent, and identifies a control factor based on the state factor with the most frequent action edge. Call this factor $i$. Then, the criteria is simply checking if the control factor has an edge with the target factor $j$, and rejecting trajectories where the control factor does not interact with the target factor. Formally, $\iota := \sum_{\mathbb{B}^{(t)} \in \bar{\mathbb{B}}} \mathbb{B}_{ji}^{(t)} = 0$.
   This requires, after identifying the control factor, a single interaction test. However, it is stringent, since it is not required that the control factor directly connects to the target object to exert control. In practice, we identify the control factor by learning $2n$ models, one for each state factor, $f^{\text{passive}}(\mathbf{s}_k) \to \mathbb{P}(S'_k | S_k = \mathbf{s}_k)$ and $f^{\text{action}}(\mathbf{s}_k, \mathbf{a}) \to \mathbb{P}(S'_k | S_k = \mathbf{s}_k, A = \mathbf{a})$. Then we identify the factor for which adding $\mathbf{a}$ helps the most.

The above strategies can reduce some sensitivity to false positives since they only require inference on a reduced subset of edges. For the same reason, they reduce the computational cost. However, they do not identify the degree of control. We utilize a simple strategy of only keeping a trajectory if the number of interactions according to the testing strategy exceeds a certain count. In other words, there should be at least some amount of interactions if $\iota$ is to keep a trajectory. The meaning of frequent is based on an interaction count $b$, which is defined differently for each method.

1. **Action Graph**: define the interaction count as the number of unique paths:
$$b := \#\text{unique paths from actions to } j$$
. A unique path is any path that contains at least one unique edge.

2. **Non-Passive**: define the interaction count as the number of non-passive edges:
$$b := \sum_{B_{ij}^{(t)} \in \bar{\mathbb{B}}} \sum_{i=0}^{n} \sum_{j=0}^{n} \mathbb{B}_{ji}^{(t)}.$$

3. **Non-Passive Target**: define the interaction count as the number of non-passive edges to the target:
$$b := \sum_{B_{ij}^{(t)} \in \bar{\mathbb{B}}} \sum_{i=0}^{n} \mathbb{B}_{ji}^{(t)}.$$

4. **Control-target**: define the interaction count as the number of control-target edges:
$$b := \sum_{B^{(t)} \in \bar{\mathbb{B}}} \mathbb{B}_{ji}^{(t)}.$$

Then $\iota := b < n_{\text{min interaction number}}$

In practice, we use action-graph interactions in all domains except Spriteworld Obstacles, Robosuite Pushing Obstacles, and Air Hockey Obstacles. In these domains, we use control-target interactions for experimental inference.

# F    INVESTIGATING $\epsilon_{\text{NULL}}$

The $\epsilon_{\text{null}}$ parameter can have significant effects on the success of the algorithm, since if selected to be too small, this will overestimate the prevalence of interactions, and if too large, interactions will fail

to be identified. While in the experiments for this work we found that a fixed $\epsilon_{\text{null}}$ is sufficient, as we see in Table 2, we provide the following strategies for identifying this parameter using information from learning the null model.

First, learn a null forward model $f : \mathcal{S} \times \mathcal{A} \times \mathcal{B} \to P(\cdot | S, A)$ that uses information about all input states to predict a distribution over $S_i'$, as in Section 4. Then compute the difference in value from the null operation on the dataset:

$$\text{diff}(\mathbf{s}, \mathbf{a}, \mathbf{s}_j, \theta)_{\text{ji}} = \log f(\mathbf{s}, \mathbf{a}, \mathbb{B}(\mathbf{v}); \theta)_j[\mathbf{s}_j] - f(\mathbf{s}, \mathbf{a}, \mathbb{B}(\mathbf{v}) \circ S_i; \theta)_j[\mathbf{s}_j] \tag{8}$$

Now, our objective is to identify the differences that most likely correspond to interactions. Non-interactions will generally have low error, since nulling should have no effect on the outcome $\text{diff}(\mathbf{s}, \mathbf{a}, \mathbf{s}_j, \theta)$. Thus, there should be a cluster of low likelihood difference outcomes. Since interactions mean that the non-nulled evaluation will be higher likelihood, $\text{diff}()$ will be higher for these values, corresponding to the second cluster. In practice, we can apply a 2-mode clustering algorithm to get the interaction and non-interaction clusters, and take the midpoint (or some other in-between point) of the mean of the two clusters.

## G    NETWORK ARCHITECTURES

In this section, we describe the network architectures used in this work. The same architecture is used for $h, f$, the interaction and forward dynamics networks.

### G.1    POINTNET ARCHITECTURES

Pointnet (Qi et al., 2017) utilizes a 1D convolution-based architecture and an order-invariance reduce function. In this work, we utilize a multilayer pointnet, which can re-append an embedding output by the final layer as an input. Our Pointnet uses a 1D convolution to embed the states and action $\mathbf{s}_1, \ldots \mathbf{s}_n, \mathbf{a}$, masks them with $\mathbf{v}$, and then follows that with a second 1D convolution. This architecture is visualized in Figure 8.

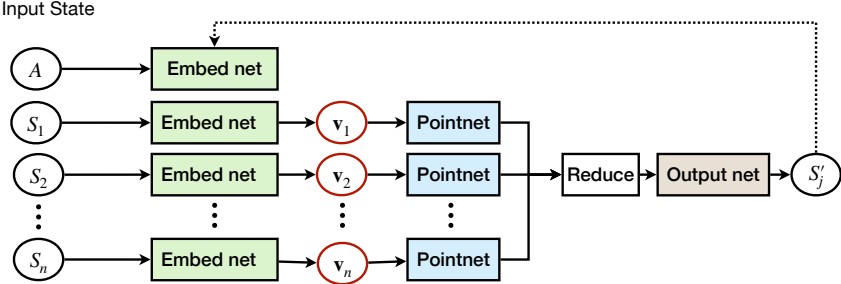

Figure 8: The Pointnet-based architecture used for interactions. Multilayer methods repeatedly re-append the output to the embeddings. Shared colors (green, yellow) denote weight sharing through 1D convolution.

### G.2    GRAPH ARCHITECTURES

We use graph neural networks (GNNs) to model the dynamics, while 1D convolutions are applied to embed the state and actions. For message passing on GNN, we utilize GCNConv layers (Kipf & Welling, 2017), and when nulling out objects, we directly remove the corresponding nodes and its edges from the graph. The architecture is visualized in Figure 9.

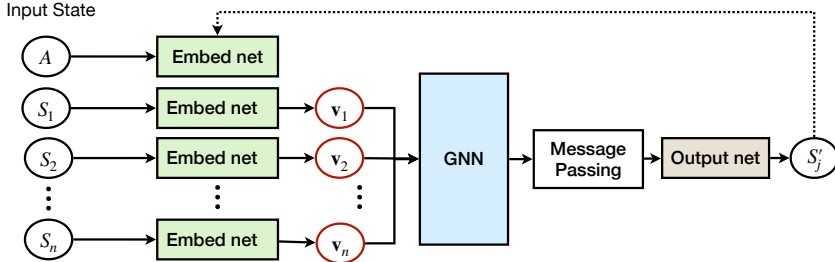

Figure 9: The GNN architecture used for interactions. Similarly, shared colors (green, yellow) denote weight sharing through 1D convolution.

### G.3 Transformer Architectures

To adapt to the transformer backbone (Vaswani, 2017), we model the dynamics using a transformer and nullify the effects of objects by setting their interactions to zero in the cross-attention maps. However, a challenge occurs when using multiple cross-attention layers: nulling out attention in one layer does not prevent information exchange in subsequent layers. Results from Random DAG domains (Table 3) also indicate that transformers perform worse than PointNet or GNN, which we will address in future work.

| Method | NCII w/ Point | NCII w/ Graph | NCII w/ Transformer |
|---|---|---|---|
| 1-in | $0.8 \pm 0.2$ | $1.0 \pm 0.1$ | $2.6 \pm 0.3$ |
| 2-in | $1.4 \pm 0.2$ | $1.4 \pm 0.2$ | $4.9 \pm 1.7$ |
| 6-in | $25.1 \pm 5.8$ | $26.3 \pm 5.2$ | $41.3 \pm 8.2$ |

Table 3: Accuracy of inference in evaluated domains from states using NCII with different backbones. Interactions are reweighted to be 50% of the test dataset.

## H Environment Details

### H.1 Spriteworld

The three 2D Spriteworld domains consist of a target ball being pushed to a goal region by a control ball in a low friction, gravity-free 7x7 meter region. The action space consists of forces upon the control ball, and the goal space is a 1m diameter circle. The reward function is sparse, with 0 reward outside and 1 reward inside the goal region. Because controlling the control ball is already challenging, and when struck the target ball is likely to pass through rather than stay in the goal region, this task can take a very substantial number of interactions to train. In the variants, we add 2 obstacles of similar mass to the target ball which impede it from reaching the goal. Ground truth interactions in this domain consist of contacts between the different objects.

The three variants differ according to the radius of the goal and target objects, and the presence of obstacles. In the Default domain, the radius of the control and target objects are $0.5$ meters, while in the **small** and **Obstacles** domains they are $0.3$ meters. In the obstacles domain are two additional obstacles, a triangle and a circle, of similar mass to the target object and where the radius of the polygon (distance from centroid to further vertex) and circle is $0.5$ meters. Each domain uses 100 time steps for a trajectory before timing out.

### H.2 Robosuite

The three Robosuite domains involve pushing a block to a desired target location using end effector control of a PANDAs robotic arm. The workspace is a $0.6 \times 0.6 \times 0.3$ meter region in length, width height respectively. The action space consists of desired end effector deltas, using an OSC controller to achieve the desired delta position. The goal space is a $0.05$m diameter circle. Again, the reward function is sparse, with 0 reward outside and 1 reward inside the goal region. Increasing the dimensionality by moving the gripper in 3D space means that interactions are even less frequent,

even though the task is over a much smaller area. In the **obstacle** variant we add 2 immovable 0.05m blocks to the domain, initialized randomly such that they do not lie on top of the goal. Once again, Ground truth interactions in this domain consist of contacts between the different objects.

The three variants differ according to the size of the target block, and the presence of obstacles. In the **Default** domain, the cube side length of the target block is 0.015 meters, while in the **small** and **obstacles** domain it is 0.007 meters, where the obstacle domain has the two additional 0.05m obstacles. Each domain uses 100 time steps for a trajectory before timing out.

### H.3 AIR HOCKEY

The three 2D Air Hockey domains consist of a puck being struck into a goal region using a paddle in a low friction 2x1 meter region, where there is gravity pulling the puck down towards the paddle. The action space consists of forces upon the paddle, and the goal space is a 0.2m diameter circle. In this domain only, we use a shaped reward instead of a sparse one, although hindsight still proves to be useful because of the complexity of the dynamics. The reward function is densified with a l2 distance, which was necessary for any policy to learn in this domain. The agent receives $-\frac{1}{2}\|s_{puck} - g\|_2^2$ reward outside and 1 reward inside the goal region. The challenge in this domain is to use a sparse interaction to achieve a downstream effect, with the puck constantly moving in and out of the goal. In the variants, we add 2 blocks of high mass to the target ball which can impede the puck from reaching the goal. Ground truth interactions in this domain consist of contacts between the different objects.

The three variants differ according to the radius of the puck and paddle, and the presence of obstacles. In the **Default** domain, the radius of the puck is 0.03m and puck is 0.05 meters, while in the **small** and **obstacle** domain the puck is 0.02 meters and the puck is 0.03 meters. The obstacle domain additionally adds three random obstacles. The goal in all domains is 0.15m radius. Each domain uses 100 time steps for a trajectory before timing out.

### H.4 FRANKA KITCHEN

In the Franka Kitchen domain, a robot with 9 degrees of freedom is tasked with controlling various kitchen objects, including the top and bottom burners, light switches, sliding and hinged cabinets, kettle, and microwave. Each object corresponds to a specific sub-task, and the robot must perform a sequence of tasks, where each is associated with the goal positions of its joints. The sparse reward is computed based on the number of completed tasks. Here in the **Default** domain, the robot's task is to open the microwave (three objects: desk and microwave). In the **Obstacle** domain, the robot's task is to open the microwave and the sliding cabinet, where the additional objects such as the kettle and burner switches are added as obstacles.

## I TRAINING DETAILS

In this section we describe the hyperparameters and training details for NCII and HInt.

All null experiments were collected with 10 seeds between 0-9. All RL experiments used 5 seeds between 0-4. The experiments were conducted on machines of the following configurations:

- 4×Nvidia A40 GPU; 8×Intel(R) Xeon(R) Gold 6342 CPU @2.80GHz
- 4×Quadro RTX 6000 GPU; 4×Intel(R) Xeon(R) Gold 6342 CPU @2.80GHz
- 4×Nvidia 4090 GPU; 8×Intel(R) Xeon(R) Gold 6342 CPU @2.80GHz
- 2×Nvidia A100 GPU; 8×Intel(R) Xeon(R) Gold 6342 CPU @2.80GHz

| Encoding Dim | 512 |
|---|---|
| Hidden | $3 \times 512$ |
| Activation | Leaky ReLu |

Table 4: Forward/inference Model

| Parameter | Value |
|---|---|
| $\epsilon_{\text{null}}$ | 1 (log-likelihood space) |
| Minimum Normalized distribution variance | 0.001 |
| Distribution | Diagonal Gaussian |
| Learning Rate | $1 \times 10^{-4}$ |

Table 5: Null Parameters

| Parameter | Value |
|---|---|
| Algorithm | DDPG |
| Batch Size | 1024 |
| Optimizer | Adam |
| Actor/critic learning rate | $1 \times 10^{-4}$ |
| Exploration Noise | 0.1 |
| $\gamma$ | 0.9 |
| Hidden Layers | $2 \times 512$ |
| $\tau$ | 0.005 |

Table 6: Reinforcement Learning Parameters

| Domain | Timeout | Normalized Goal Epsilon | Null Train Steps | RL Train Steps |
|---|---|---|---|---|
| Spriteworld Default | 100 | 0.1 | 1M | 50M |
| Spriteworld Small | 100 | 0.15 | 1M | 50M |
| Spriteworld Obstacles | 100 | 0.2 | - | 50M |
| Spriteworld Velocity | 100 | 0.2 | - | 50M |
| Robosuite default | 100 | 0.15 | 1M | 10M |
| Robosuite small | 100 | 0.15 | - | 10M |
| Robosuite obstacles | 100 | 0.15 | - | 15M |
| Air Hockey default | 400 | 0.2 | 1M | 10M |
| Air Hockey small | 400 | 0.2 | - | 20M |
| Air Hockey obstacles | 400 | 0.2 | - | 20M |
| Franka Kitchen default | 200 | 0.2 | 2M | 3M |

Table 7: Domain Specific Parameters

| Domain | HInt learned | HInt | Hind | Prio | FPG | Vanilla | ELDEN | CAI |
|---|---|---|---|---|---|---|---|---|
| Sprites (50M) | 62.21 | 22.10 | 23.27 | 26.02 | 17.15 | 22.57 | 40.12 | 69.34 |
| Air (10M) | 12.50 | 4.33 | 4.59 | 6.05 | 5.24 | 3.31 | 9.75 | 17.23 |
| Robo (10M) | 21.17 | 11.52 | 15.32 | 11.28 | 8.59 | 8.21 | 18.28 | 25.20 |
| Franka (3M) | 16.68 | 9.26 | 8.45 | 8.92 | 10.39 | 9.16 | 13.25 | 22.48 |

Table 8: Wall Clock Compute Time in Hours

## J HINDSIGHT SAMPLING ABLATION

In this work, we focused on the introduction of interactions into Hindsight Filtering using the "final" sampling scheme, which takes the last state of a trajectory as the hindsight goal. In practice, different sampling schemes can be used for hindsight Andrychowicz et al. (2017), including sampling any state after the one observed "future" or any state from the trajectory "episode." We provide some analysis

| Method | NCII w/ Point | JACI | Gradient | Attention | NCD |
|---|---|---|---|---|---|
| 1-in-nonlinear | $\mathbf{0.9 \pm 0.2}$ | $2.4 \pm 0.8$ | $32.5 \pm 6.1$ | $37.4 \pm 0.7$ | $21.2 \pm 1.1$ |
| 2-in-nonlinear | $\mathbf{2.3 \pm 0.1}$ | $\mathbf{2.5 \pm 0.2}$ | $36.4 \pm 0.4$ | $21.8 \pm 0.9$ | $19.8 \pm 2.0$ |
| 40-dim | $\mathbf{1.4 \pm 0.1}$ | $2.4 \pm 0.5$ | $34.7 \pm 4.4$ | $26.4 \pm 6.9$ | $12.5 \pm 0.8$ |

Table 9: Misprediction rate (lower is better) of inference in additional domains from state, similar to Table 1. Interactions are reweighted to be 50% of the test dataset. Boldface indicates within $\sim 1$ combined standard deviation of the best result. k-in-nonlinear uses nonlinearities in the random DAG instead of a linear relationship. 40-dimensional uses a 40 dimensional state.

suggesting that hindsight filtering is applicable in any sampling scheme in Figure 10. Note again that in general, we used the "final" sampling strategy for all other experiments (Figure 4, Figure 7, Figure 12).

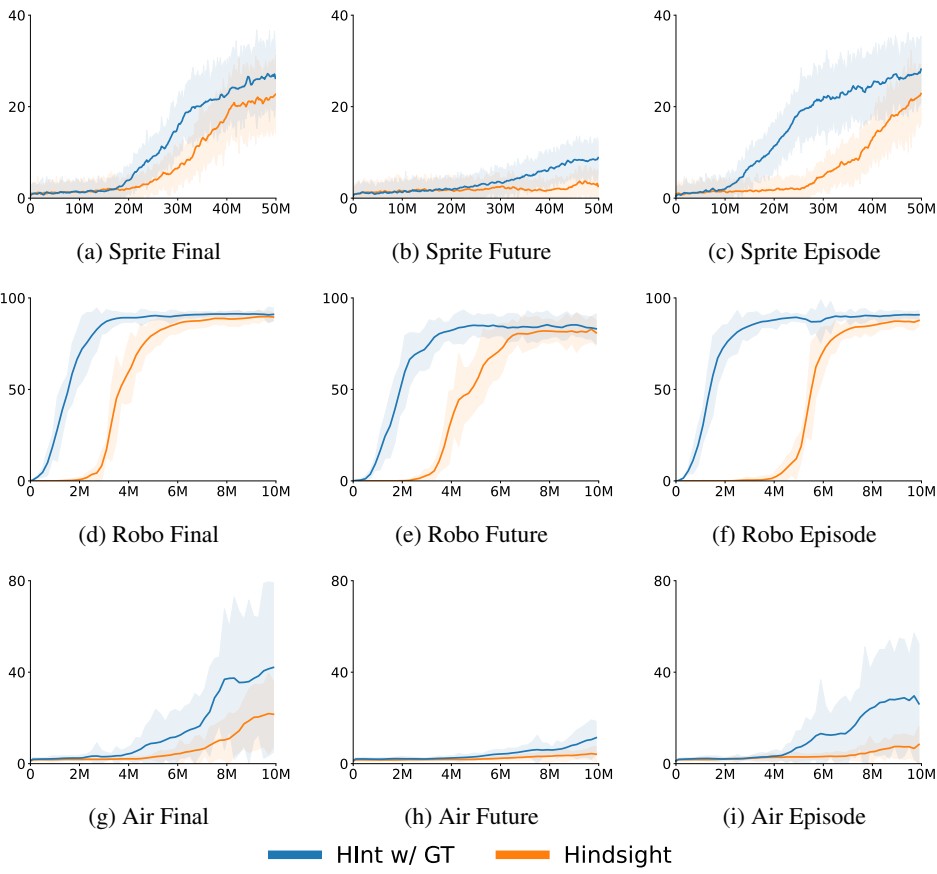

Figure 10: Comparison of HInt and Hindsight using different HER sampling schemes "final", "future" and "episode", with 3 trials for each sampling scheme. In these experiments, we modified the sampling scheme for both HInt and Hindsight to the sampling strategy indicated in the caption. The shading indicates standard error. Sprite-, Air- and Robo- domains each use the default variant. The Y axis is average reward per episode.

## K  VISION EXPERIMENTS

While the objective of this work is to demonstrate a generally applicable method for utilizing counterfactual nulls for interaction inference (NCII) and interaction filtering for GCRL (HInt), we briefly explore the scaling capabilities of NCIIand HInt. In both interaction inference and GCRL, performance in higher dimensional states remains a challenging problem. While our results certainly

| Method | NCII w/ Point | JACI | Gradient | Attention | NCD |
|---|---|---|---|---|---|
| Sprite-1-vision | $\mathbf{13.9 \pm 0.5}$ | $18.1 \pm 0.2$ | $26.4 \pm 2$ | $39.6 \pm 4.5$ | $21.2 \pm 1.1$ |

Table 10: Misprediction rate (lower is better) of inference in Spriteworld-1 domain. Interactions are reweighted to be 50% of the test dataset. Boldface indicates within $\sim 1$ combined standard deviation of the best result.

suggest that the NCIIis competitive with baselines in both domains, these scaling questions remain unsolved.

Empirically, we take a segmentation of each object in the frame (given from the simulator) in the Sprite default domain, and train a variational autoencoder Kingma (2013) on a frame stack of 3 frames of each segmented object with a latent dimension of 128. We then append object-centric features pixel position and velocity, duplicated to 128 dimensions. This is then used as the input for both NCIIand RL using HInt. We first compare NCIIto interaction inference baselines in the Spriteworld Default domain in Table 10. Then we provide a performance curve for HIntwhen compared against the Hindsight and Vanilla RL baselines in Figure 11.

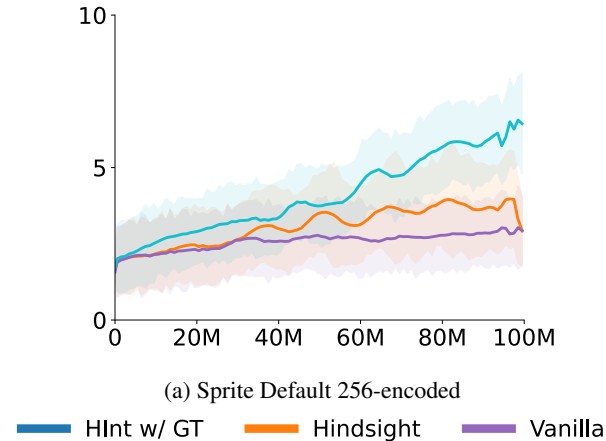

(a) Sprite Default 256-encoded

HInt w/ GT     Hindsight     Vanilla

Figure 11: Comparison of HInt, Hindsight and Vanilla RL, 3 trials for each, on Spriteworld Default using 256-dimension image encodings. The shading indicates standard error. The Y-axis is average reward per episode.

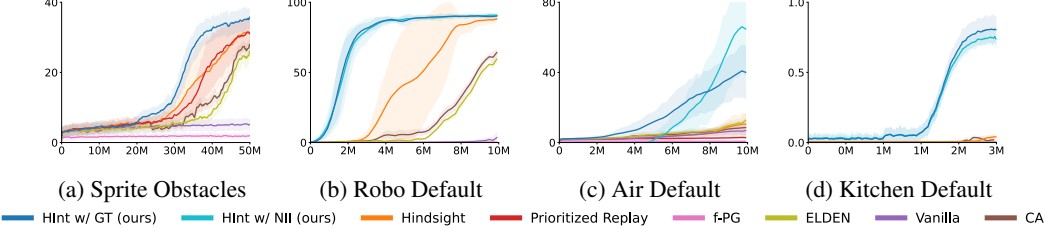

(a) Sprite Obstacles     (b) Robo Default     (c) Air Default     (d) Kitchen Default

HInt w/ GT (ours)   HInt w/ NII (ours)   Hindsight   Prioritized Replay   f-PG   ELDEN   Vanilla   CAI

Figure 12: Addition of Causal Action Influence (CAI) Seitzer et al. (2021) baseline in selected domains. This baseline is similar to ELDEN, but has more inductive bias towards actions. The Y axis is the average reward per episode.

## L   ADDITIONAL COMPARISONS

We also compare with **Causal action influence (CAI)** (Seitzer et al., 2021). CAI uses conditional mutual information to infer the local causal relations, detecting when and what the agent can influence the state variables with its actions. Then they employ the influence as the intrinsic motivation for

exploration that benefits sample efficiency. The results in selected domains are given in Figure 12, where CAI performs similarly to ELDEN empirically.

## M ROLLOUT VISUALIZATIONS

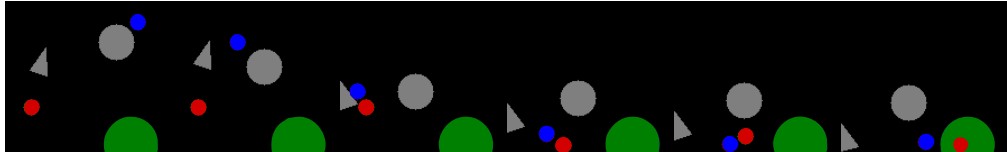

Figure 13: Selected frames from a successful policy rollout for Spriteworld Obstacles

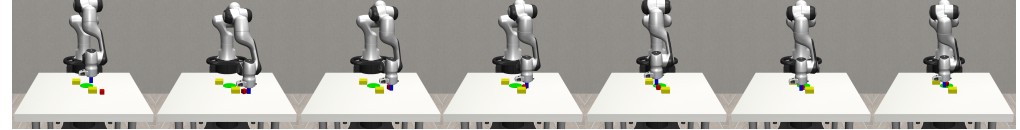

Figure 14: Selected frames from a successful policy rollout for Robosuite Obstacles

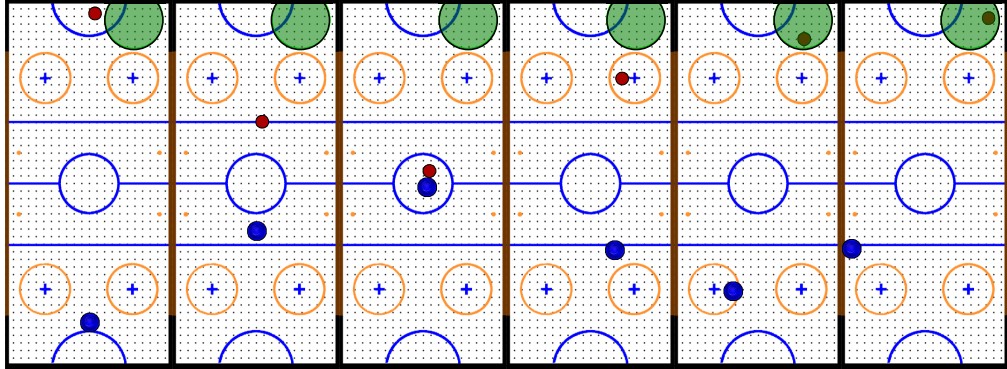

Figure 15: Selected frames from a successful policy rollout for Air Hockey Default

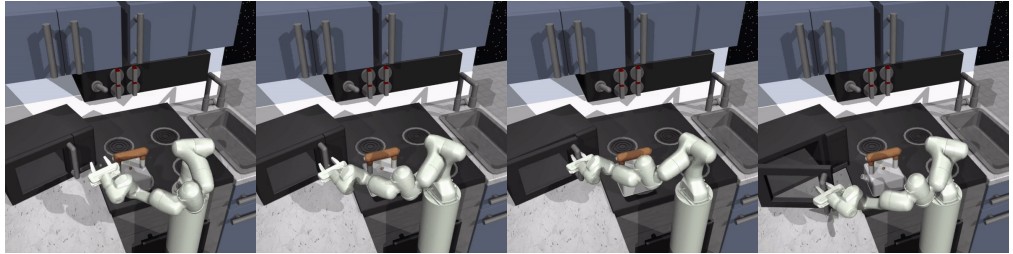

Figure 16: Selected frames from a successful policy rollout for Franka Kitchen

