# OpenReview forum: "Null Counterfactual Factor Interactions for Goal-Conditioned Reinforcement Learning"
_ICLR.cc/2025/Conference — ICLR 2025 Poster_

### Official Review · Reviewer_SPH4 · 2024-10-25

**Soundness:** 3
**Presentation:** 3
**Contribution:** 3
**Rating:** 8
**Confidence:** 3

**Summary:**

The paper proposes Hindsight relabeling using Interactions (HInt), a variant of Hindsight Experience Replay (HER) that leverages interactions to filter candidate trajectories for relabeling. Drawing inspiration from causality, an interaction is defined as an instance where removing (or nulling) an object would have an impact on the next state of another object (Null Counterfactual Interaction Inference or NCII). Given object-centric representations, the paper proposes to learn a masked dynamics model which can predict the next state of an object conditioned on what other objects are present. An influence of object A on object B is then detected by comparing the difference of the predictions for B with and without A against a threshold. During training, interaction detection is amortized in an interaction classifier. The main proposed criterion for using a trajectory for HER is the detection of a path in the unrolled graph corresponding to interactions, leading from an action to a target object (hence, an influence of the action on the target object). Experiments in two abstract and three robotics-inspired continuous control domains show increased sample efficiency when using HInt. An analysis suggests that HInt filters out trajectories in which the target object does not move (in the Spriteworld domain).

**Strengths:**

The argument for an interaction-based inductive bias in HER is well motivated. Moreover, the interpretation of a deviating transition probability under a null counterfactual as an interaction between objects is intuitive and concise. The existence of a path from an action to the target state as a filtering criterion for HER is well founded in causality and illustrated well by figure 2.

The domains considered for the experimental evaluation are relevant and sufficiently established. Table 1 indicates that NCII is more accurate in detecting interactions than the considered baselines. The RL performance is demonstrated to benefit significantly from using HInt.

The writing in the main text is clear and the presentation is well structured.

**Weaknesses:**

In my opinion, moving too much crucial algorithm components to the appendix is a main weakness of the paper. The main text conveys the impression that it presents the whole algorithm except for some unimportant implementation details, and that this algorithm achieves good performance in practice. However, the content of Appendix C and especially Appendix D seem to be quite important, and are probably crucial for obtaining the empirical results that were reported.

In particular the filtering criteria presented in Appendix D deviate from the intuitive path-from-action-to-target definition in the main text. Moreover, a somewhat simplified and engineered criterion is then used for the experiments. Yet, it is only referred to as one of several “alternatives” in the main text (line 318). In my opinion, it should be made more clear what form of the algorithm is actually used for the experiments and which components are crucial for obtaining good performance. An ablation study with different filtering criteria would be interesting, for example.

My understanding, based on the last paragraph of appendix D, is furthermore that for the experiments, only interactions between the controlled object and the target object were detected and used as a criterion. This is a much simpler algorithm than what is presented in the main text and effectively uses domain knowledge (as it happens to be sufficient to consider such interactions in the chosen domains). Moreover, another hyperparameter thresholding the interaction frequency in a trajectory is introduced. Combined, this makes me question the claim that NCII is really more general than using heuristics (line 87).
As the algorithm used in the experiments is considerably simplified, it seems like running CAI [1] as a baseline is required. CAI simply estimates the influence of actions on the target object. It would be interesting to see how much HInt can add in terms of sample efficiency to this approach.

The content of Appendix C reads like quite a few tricks were needed to get HInt to work well. In particular the reweighing based on the log probability of a transition seems important and should therefore be mentioned in the main text.

The writing in Appendix D is sometimes a bit dense and hard to understand, for example the enumeration point “1. Non-passive”. I think there is potential for illustrating these strategies better.

[1] Seitzer, Maximilian, Bernhard Schölkopf, and Georg Martius. "Causal influence detection for improving efficiency in reinforcement learning." Advances in Neural Information Processing Systems 34 (2021): 22905-22918.

**Questions:**

* It would be interesting to see wall-clock time comparisons with the baselines as HInt adds quite a bit of complexity to them.

* I would have expected an expectation over the goal in the RL objective in like 181.

* The next paragraph (starting from line 183) is written like the goal space is equal to the state space. However, in the rest of the paper this is not the case.

* ‘min’ in equation (2) should be ‘max’.

* Why is no absolute value taken in equation (3) when thresholding the difference of log probabilities.

* In line 303, the filtering function is defined is defined as a decision to reject a trajectory while in appendix D it seems to be the decision to accept a trajectory.

* I think (left) and (right) are switched in the caption of Figure 1.

---

> ### Author Response · Authors · 2024-11-19
> **Response to Reviewer SPH4**
>
> We thank you for the encouraging and insightful comments. All of them are invaluable for further improving our manuscript. Please refer to our response below.
>
> > Q1: In my opinion, moving too much crucial algorithm components to the appendix is a main weakness of the paper...
>
> **R1:** While we agree that the appendix provides valuable details on the implementation of the algorithm, we do not want to make it appear as though key implementation details were buried in the appendix. To clarify, we used the action graph implementation of HInt in all the domains except for: Sprite Obstacles, Air Obstacles, and Robo Obstacles. In these domains, we found that using the control method resulted in more stable performance. We modify the main paper to include additional clarification in the methods section of the distinction between the control-target filtering strategy, and in the experiments section to clarify which domains utilized the which strategy.
>
> In practice, the control method could be applied interchangeably with the action graph formulation in other environments, since it is essentially an action graph formulation where graphs with more than two state factors in the control chain are filtered out. As a result, we do not consider the ``control'' method to be a heuristic injecting significant amounts of knowledge about the environment since it still relies on interaction identification to 1) identify the control object through correlation with actions 2) identify interactions with the target object using the null assumption. The other alternative filtering schemes are simply strategies which we employed and we believe would be informative to a researcher building on this method.
>
> Interaction identification is a different problem from just action influence detection, as it relies on identifying the counterfactual possibilities from relations that are often quite sparse. Inductive biases about interactions often have to be encoded through physical assumptions about the simulation, as seen in [1]. Furthermore, the problem of causal interactions is similar to that of actual cause [2,3], especially functional actual cause [4], and remains an open and challenging problem for reasoning. NCII, while it makes some assumptions, is more general than just a contact heuristic because it does not rely on special knowledge about the physical domain such as contact or physics, except for the fact that state factors can be not present and thus "nulled" out. **To clarify this point, however, we are currently running CAI on several of our domains, with upcoming results to be added to this thread, to demonstrate that the benefit of the algorithm is not purely heuristic.**
>
> *Related revised parts: Section 4.2, Appendix D*
>
> > Q2: The content of Appendix C reads like quite a few tricks were needed to get HInt to work well.
>
> **R2:** With respect to the heuristic elements in Appendix C, while this passive reweighting strategy is meaningful for empirical results, it is a common practice in interaction-based methods because in many domains, especially when dealing with random actions, interactions are exceedingly sparse. In particular, we note that other work involving interactions such as [4, 5] also employ similar strategies when learning the interaction models. While we entirely agree that these are important implementation details, they are 1) specific to domains with sparse interactions 2) Not a core contribution of this particular work. For clarity, we added the following to the methods section:
>
> In practice, learning to identify interactions is challenging when interactions are rare events, such as a robotic manipulation domain where the robot rarely touches the object it should manipulate. These challenges have been approached in prior work [4,5] by reweighting and augmenting the dataset using statistical error. We include a description of how those methods are adapted and used in this algorithm in Appendix C.
>
> We do not convey that the work in the main paper is entirely divorced from the details in the appendix, or that those details are entirely unimportant, but rather just that those techniques have been employed in similar forms in prior work.
>
> *Related revised parts: Section 4.1, Appendix C*

---

> > ### Author Response · Authors · 2024-11-19
> > **Response to Reviewer SPH4**
> >
> > > Q3: It would be interesting to see wall-clock time comparisons with the baselines as HInt adds quite a bit of complexity to them.
> >
> > **R3:** We will provide wall-clock comparisons on the spriteworld domains, but the cost of HInt is comparable or better than other model-based learning methods such as ELDEN or CAI, but more expensive than vanilla hindsight.
> >
> > > Q4: I would have expected an expectation over the goal in the RL objective in like 181.
> >
> > **R4:** This is a good point we are maximizing the expectation over the goal, transition dynamics, and policy. The equation on 181 is just an expression for the return with respect to a particular state and goal. We’ve added additional clarification to address this.
> >
> > *Related revised parts: Lines 182-183, Section 3.1*
> >
> >
> > > Q5: The next paragraph (starting from line 183) is written like the goal space is equal to the state space. However, in the rest of the paper, this is not the case.
> >
> > **R5:** We added the following clarification to the beginning of the paragraph: Note that in this setting, we operate under the formulation where the goal space need not be equal to the state space, for example, the state of a particular state factor.
> >
> > *Related revised parts: Lines 185-187, Section 3.1*
> >
> > > Q6: ‘min’ in equation (2) should be ‘max’.
> > We have modified equation 2 with max in the revision.
> >
> > **R6:** Why is no absolute value taken in equation (3) when thresholding the difference of log probabilities.
> > In practice, we are interested in how much higher the likelihood of the actual outcome compared with the null counterfactual. If we used absolute value, then it might be the case that the null counterfactual is actually higher likelihood, but this would not indicate an interaction
> >
> > > Q7: In line 303, the filtering function is defined is defined as a decision to reject a trajectory while in appendix D it seems to be the decision to accept a trajectory.
> >
> > **R7:** We modified the language of the appendix to use the language of rejecting a trajectory in each of the contexts.
> >
> >
> > > Q8: I think (left) and (right) are switched in the caption of Figure 1.
> >
> > **R8:** Thank you for this catch, we have switched left and right in the revision.
> >
> > [1] Kang, Bingyi, et al. "How Far is Video Generation from World Model: A Physical Law Perspective." arXiv preprint arXiv:2411.02385 (2024).
> >
> > [2] Halpern, Joseph Y. Actual causality. MiT Press, 2016.
> >
> > [3] Beckers, Sander. "Causal explanations and XAI." Conference on causal learning and reasoning. PMLR, 2022.
> >
> > [4] Chuck, Caleb, et al. "Automated Discovery of Functional Actual Causes in Complex Environments." arXiv preprint arXiv:2404.10883 (2024).
> >
> > [5] Chuck, Caleb, et al. "Granger Causal Interaction Skill Chains." Transactions on Machine Learning Research.

---

> > > ### Comment · Reviewer_SPH4 · 2024-11-22
> > > **Response**
> > >
> > > Thank you for your detailed response to my questions and concerns. It was indeed quite helpful in clearing up some ambiguities.
> > >
> > > Q1/R1: It was not clear from reading the original submission that the action graph variant was also used in the experiments. Thank you for clarifying this and adding an explanation in the main text. I still think that the sentence “In practice, we use the control-target interactions for experimental inference.” in line 845 in Appendix D is misleading as it sounds like the full action graph is never used in the experiments. Would it be possible to change that formulation?
> > >
> > > Q2/R2: Thank you for mentioning the techniques from Appendix C in the main text. I believe this is provides useful guidance to the reader that is interested in the implementation of the method.
> > >
> > > I furthermore appreciate the effort you are putting into running CAI and providing wall-clock comparisons. My other questions and remarks were answered or sufficiently addressed. Thank you!
> > >
> > > I have increased my score based on the revision and your response.

---

> ### Author Response · Authors · 2024-11-25
> **Updates on CAI results, wall-clock time comparison, and ablations with different filtering criteria**
>
> Thank you for acknowledging that we have addressed your concerns and raising the score.
>
> To address your further concerns on Q1/R1, we have updated Appendix D accordingly. Specifically, we use action-graph interactions in all domains except Spriteworld Obstacles, Robosuite, Pushing Obstacles, and Air Hockey Obstacles. In these domains, we use control-target interactions for experimental inference.
>
> We would also like to update you with the additional experimental results below.
>
> ### **CAI Results**
>
> Reulsts are given in **Figure 12, Appendix K**. We evaluated CAI across four domains: Sprites Obstacles, Robo Default, Air Default, and Franka Default. In these domains, CAI performs comparably to ELDEN, as both methods incorporate an inductive bias towards interactions, with CAI specifically emphasizing an inductive bias towards actions.
>
> ---
> ### **Wall-clock time comparison**
>
> We provided the wall-clock time comparison in **Table 8** in the revised appendix, showing that HInt does not require significantly more computational time compared to the baselines. While HInt-learned takes more time, it still outperforms CAI, which also infers causal structures among objects. Note that the wall-clock time reported for ELDEN excludes the dependency inference component, making it appear more efficient than HInt-learned. Overall, our approach does not bring substantial computational overhead compared to other interaction-based methods.
>
> ---
>
> ### **Ablations with different filtering criteria**
>
> We have compared HInt and Hindsight on different sampling schemes ("final", "future", "episode"). Results are given in the revised **Figure 10** and **Appendix I**, which suggest that hindsight filtering is applicable in any sampling scheme.
>
> Please do not hesitate to let us know if you have any additional comments or concerns. Thank you again for your valuable feedback and effort!

---

> ### Comment · Reviewer_SPH4 · 2024-11-26
> **Response to additional updates**
>
> Thank you for fixing the formulation in appendix D, for the CAI results, and wall clock times. I appreciate the effort you put into improving the paper during the rebuttal!
>
> I agree that the wall clock time of HInt is comparable to the baselines and therefore reasonable. The comparison to CAI indeed shows a significant benefit of filtering based on interactions instead of action influence alone.
>
> I just noticed that in Figure 4, NCII  is sometimes referred to as NII or NCI. It would be better to be consistent here.
>
> Thank you for the new results in Appendix I. From the caption of Figure 10, it is not entirely clear to me, which sampling scheme was used for HER. Maybe this could be clarified with an additional sentence.
>
> As my main concerns have been addressed and the remaining small issues can be addressed easily, I have raised my score.

---

> > ### Author Response · Authors · 2024-11-26
> > **Clarifications and improvements to Figure 4,10**
> >
> > We greatly appreciate the reviewer's effort and prompt responses in improving the quality of this work! We have updated the paper with a revision that replaces instances of NCI and NII with NCII. We have also updated the caption in Figure 10 to indicate that the sampling scheme was modified for both HER and HInt.
> >
> > Thank you again for you positive feedback!

---

### Official Review · Reviewer_xepm · 2024-11-01

**Soundness:** 3
**Presentation:** 3
**Contribution:** 3
**Rating:** 8
**Confidence:** 4

**Summary:**

In the context of goal-conditioned RL, building on top of Hindsight Experience Replay, the paper proposes a filtering method that aims to improve the efficiency of learning. Under the proposed definition of interaction that is based on the change of the transition probabilities under null counterfactuals, a masked forward dynamic model is learned to identify interaction (NCII). Then the method filters the trajectory to be relabeled and only keeps those that the agent interacted with the target (NInt). The effectiveness of NCII and the improvements of NInt are verified by empirical analysis on simulated environments compared with established methods.

**Strengths:**

The problem setup is well motivated and the proposed algorithm extends , HER, an important technique in goal conditioned RL, to settings where it doesn’t work well and is effective. The presentation from the background of HER to the proposed method is smooth and well thought out, except a few minor places that can use some polish.

**Weaknesses:**

1. The null operation in Equation 3 depends on the threshold \( \epsilon_{\text{null}} \). This is an important part of the algorithm. Discussion on how to choose it and an ablation on the sensitivity of this threshold would make the analysis more comprehensive. More specifically we are interested in answering the following questions (actionable feedback below):
   - How sensitive is NCII to the choice of the threshold?
   - Does one threshold work across different environments in Table 1, or does each environment variant require a different threshold?

 Figures showing how the accuracy of HCII varies corresponding to a range of thresholds for environments, or one variant from each environment the authors already considered Table 1, would be compelling. Additionally, for a few selective environments that are sensitive to thresholds in the previous ablation, how does the episode reward change when HCII with different thresholds is used in HInt? This second ablation may not be necessary if HCII is shown to be robust across a range of thresholds in the previous one. The range of thresholds should be selected by the authors to show if there are values on the left or right tail where the algorithm starts to break down and success rates start to fall off. Success rate is the metric.

2. Hindsight Experience Replay (HER) is an important baseline here. HER has several variants for how the goal is chosen, including “future,” “final,” and “episode.” It seems that, but it’s not clear, the HER implementation here refers to the default “final” variant. Expanding the baseline in Figure 4 to include other variants of HER, especially both the “final” and “future” variants, would make the comparison more comprehensive. This is particularly relevant as the performance difference between HInt and HER is small in a few environments in Figure 4, and a stronger variant of HER might change the gap here. This would entail running on the environments in Figure 4 and reporting on the already established metric, only this time under the alternative versions of HER goal selection strategies.

3. In Equation 3, it appears that the logarithm is intended to apply to the entire subtraction; however, the syntax suggests otherwise.

4. There is a typo on line 268, page 5: “using using.”

**Questions:**

1. On page 5 around line 221, how exactly does the extra action column work with the core \( n \times n \) submatrix corresponding to the states? It appears that the interaction is defined around a pair of states. I also have the same confusion with Figure 2.

2. On page 5 around line 232, the mentioning of the vector \( \mathbf{V}^k \) would need more context. It seems to be a vector to zero out a column of the interaction matrix \( \mathbb{B} \), but it is not very clear. How is it related to, and what exactly is the property that not all tasks exhibit on line 233?

3. How should we deal with cases when there are very few trajectories satisfying the interaction criterion?

4. In Table 1, it is listed as accuracy, but it seems like lower values are better, which is a bit confusing.

---

> ### Author Response · Authors · 2024-11-19
> **Response to Reviewer xepm (Part 1)**
>
> We thank the reviewer for the insightful and encouraging comments. Please see our response as follows.
>
> > Q1: The null operation in Equation 3 depends on the threshold ( \epsilon_{\text{null}} )...
>
> **R1:** We appreciate that the reviewer's identification of $\epsilon_\text{null}$ as a key parameter. In practice, we used the same null epsilon parameter of $\epsilon_\text{null} = 1$ for all environments and all experiments, even across domains such as random vectors, where the dynamics differ significantly from those in the physical interaction domains such as air hockey and SpriteWorld. This is because when the state inputs and deltas are normalized the effect of interactions as a result of an interaction is fairly significant. For example, the average change in velocity as a result of an interaction such as a ball hitting another ball or a robot manipulating an object in most physical domains is significant compared with the effect of drag. *We are also working on an ablation illustrating the effect of changing the $\epsilon_\text{null}$ value in both the random sprite domain and the random vectors domain (to show the difference across two significantly different dynamics), to illustrate the insensitivity of this hyperparameter, and will include that in this thread when those runs are completed.*
>
> This being said a poor choice of $\epsilon_\text{null}$ can result in the method failing. We suggest the following strategy for selecting $\epsilon_\text{null}$:
> We can take the null model $f(\mathbf s, \mathbf a, \mathbb B(\mathbf v))$ and observe that the interaction states will be those where the difference between the nulled and non-nulled model outputs will be larger (meaning the non-nulled model will have higher likelihood.) On the other hand, in non-interaction states, the error should be small. Thus, we can take the differences, identify two clusters, and then take the midpoint between the higher cluster center (corresponding to interaction states) and the lower cluster center (corresponding to non-interaction states. As $\epsilon_\text{null}$. We add a formalization of this hyperparameter selection strategy in the appendix to strengthen the overall generalizability of the paper and add some analysis illustrating the threshold selected by applying these operations.
>
> *Related Revised Parts: Appendix E*
>
>
> > Q2: Hindsight Experience Replay (HER) is an important baseline here
>
> **R2:** HER sampling is a vital element on which we ran early experiments. In the work, we use the "final" variant of HER, and thus compare directly against this variant. However, we also implemented the final and future version of HER, and are *presently running an ablative comparison of each of these methods for both HInt and baselines on the random Sprite environment. As these experiments are currently running, we will update this thread with the results of these experiments.*

---

> > ### Author Response · Authors · 2024-11-19
> > **Response to Reviewer xepm (Part 2)**
> >
> > Questions:
> > > Q3: On page 5 around line 221, how exactly does the extra action column work with the core ( n \times n ) submatrix corresponding to the states? It appears that the interaction is defined around a pair of states. I also have the same confusion with Figure 2.
> >
> > **R3:** In this work, we treat actions as another state factor (but for simplicity, we refer to $n$ factors instead of $n+1$ factors, and learn relationships independently). This means that the relationship between actions and state elements can also be identified. In practice, the agent will never observe "nulled" actions, which means that even though we can "null" out actions, this can result in out-of-distribution errors, though we do not observe them in practice. This design choice allows us to directly observe the effect of actions on the forward dynamics, though this can also result in conflating issues if actions "leak" information about one state factor (information about a nulled state factor is encoded through the policy in the action selection). Again we do not observe this in practice.
> >
> > > Q4: On page 5 around line 232, the mentioning of the vector ( \mathbf{V}^k ) would need more context. It seems to be a vector to zero out a column of the interaction matrix ( \mathbb{B} ), but it is not very clear. How is it related to, and what exactly is the property that not all tasks exhibit on line 233?
> >
> > **R4:** The interpretation that this vector zeros out a column of the interaction matrix is correct. This is only possible in tasks where an object is known to not be present in a trajectory. For example, a version of the random sprites environment with a triangle, circle, control, and target object, where a different random subset of the objects (ex. triangle, control target, circle control target, control target) are present in any given trajectory. In the example, a trajectory such as $triangle, control, target$, V^{k} would zero out the column corresponding to the circle. We added this clarification to this section.
> >
> > *Related Revised Parts: Section 4.1*
> >
> > > Q5: How should we deal with cases when there are very few trajectories satisfying the interaction criterion?
> >
> > **R5:** It is actually already often the case that every few trajectories satisfy interactions, which is what allows HInt to provide such a significant performance benefit. In this work we primarily allow random exploration to eventually collect enough desired interactions, though this work could be augmented in parallel with an exploration strategy that induces interactions. Alternatively, the HER buffer can be warm-started with demonstration data, which is likely to induce useful interactions. Note that in the absence of hindsight data, the agent can still sample from the replay buffer, just without hindsight (and if interactions are not happening, then this would be preferable. In practice to prevent early RL training from degenerating, we also do not sample from the HER buffer with less than 1000 samples.
> >
> > > Q6: In Table 1, it is listed as accuracy, but it seems like lower values are better, which is a bit confusing.
> >
> > **R6** This is a good catch, and we list the table values as misprediction rates rather than accuracy.
> >
> > *Related Revised Parts: Table 1*

---

> ### Author Response · Authors · 2024-11-25
> **Update on $\epsilon_\text{null}$**
>
> We again thank the reviewer for their response and constructive feedback, we have provided a general response, and also include additional details on the updated analysis of  $\epsilon_\text{null}$ here.
>
> See *General response, Table 2 in Appendix E*
>
> We ablated on the $\epsilon_\text{null}$ hyperparameter with 3 seeds for each setting of $\epsilon_\text{null}$ to empirically analyze the dependence on the threshold for identifying null interactions. Our experiments evidence a limited dependence on this parameter except at the upper and lower extremes, as values within 0.3 to 2.5 show performance within variance for both random DAG and Box2D domains, which have significantly different dynamics. Since epsilon-null compares the difference in normalized log-likelihood of the predictions, this is an invariance across two orders of magnitude. This result corroborates with the evidence that across all of the domains that we tested, air hockey, Spriteworld, Robot pushing, Franka Kitchen and random DAG, and Random DAG with nonlinear relationships, the same $\epsilon_\text{null} = 1$. We believe that this suggests the efficacy of NCII across a variety of domains, though future work can investigate strategies for automatically setting the $\epsilon_\text{null}$ parameter.
>
> Please let us know if you have any other questions. We are more than happy to discuss and address them. Thank you again for your positive feedback!

---

> > ### Comment · Reviewer_xepm · 2024-11-26
> > **All questions answered**
> >
> > All of my questions have been answered.

---

### Official Review · Reviewer_EVkv · 2024-11-04

**Soundness:** 3
**Presentation:** 3
**Contribution:** 2
**Rating:** 5
**Confidence:** 3

**Summary:**

This paper considers a notion of null counterfactual: a cause object is interacting with a target object if in a world where the cause object did not exist, the target object would have different transition dynamics. Using this definition, the paper proposes ways to simulate virtual rollouts and perform data augmentation by leveraging null counterfactual. Hindsight Experience Replay is playing a key role in the algorithm, and the algorithm seems to inject some compositionality into hindsight replay. Toy tasks and robotic tasks are considered for evaluation.

**Strengths:**

- The notion of null counterfactual is interesting.
- The paper manages to devise a full-algorithm from this notion, and showed practical gain in object-centric robotic domains.
- Goal-conditioned RL is an important area of research, and using null counterfactual for data augmentation is a promising direction.

**Weaknesses:**

- This method seems relatively limited to object-centric domains, where the dynamics between objects is relatively simple.
- Certain set-based architecture (such as PointNet and some version of GNN) might not work in general domains to model dynamics.
- The simulated nulling procedure and the filter criterion feel very heuristic and specific to the considered domains.

**Questions:**

See the weakness sections.

---

> ### Author Response · Authors · 2024-11-19
> **Response to Reviewer EVkv**
>
> We thank the reviewer for the insightful feedback, please see the following for our response.
>
> > Q1:This method seems relatively limited to object-centric domains, where the dynamics between objects is relatively simple.
>
> **R1**:  Yes, in this work, we assume the framework operates in object-centric domains. However, the null counterfactual reasoning framework can also be applied to general factored domains (i.e..., Factored MDPs), as the learning and inference processes using Equations (3) and (4) do not require the states to be explicitly factorized as objects.
>
> Additionally, we believe that learning object interactions is non-trivial, particularly in highly interactive, object-rich environments where such interactions are critical for decision-making. Hence, we believe that robotics and embodied AI that require compositional and controllable representations and decision-making systems could highly benefit from object-centric RL [1-3].
>
>
>
> [1] Hong, Yining, et al. "Multiply: A multisensory object-centric embodied large language model in 3d world." Proceedings of the IEEE/CVF Conference on Computer Vision and Pattern Recognition. 2024.
>
> [2] Shi, Junyao, et al. "Composing Pre-Trained Object-Centric Representations for Robotics From" What" and" Where" Foundation Models." arXiv preprint arXiv:2404.13474 (2024).
>
> [3] Zheng, Ying, et al. "A Survey of Embodied Learning for Object-Centric Robotic Manipulation." arXiv preprint arXiv:2408.11537 (2024).
>
> > Q2: Certain set-based architecture (such as PointNet and some version of GNN) might not work in general domains to model dynamics.
>
> **R2**: Thank you for pointing this out. We would like to emphasize that our framework is architecture-agnostic, meaning any architecture can be used to model dynamics within it. PointNet and GNN are the architectures we selected for the simulated dynamics systems and RL environments in this work. Additionally, we include results for Transformers in Appendix E.3; however, these did not perform as well empirically as the other two architectures.
>
> We also argue that PointNet and GNN hold potential for scalable applications, especially when object-factored representations are available. These representations can be learned through object-centric video encoders such as Video-DINOSAUR [4] or object-aware 3D perception models like Object-Aware Gaussian Splatting [5] and other robotic foundation models. Once the representations are obtained from these encoders, GNNs or PointNet can still be used as effective interaction models.
>
>
> [4] Zadaianchuk, Andrii, Maximilian Seitzer, and Georg Martius. "Object-centric learning for real-world videos by predicting temporal feature similarities." Advances in Neural Information Processing Systems 36 (2023).
>
> [5] Li, Yulong, and Deepak Pathak. "Object-Aware Gaussian Splatting for Robotic Manipulation." ICRA 2024 Workshop on 3D Visual Representations for Robot Manipulation.
>
>
>
> > Q3: The simulated nulling procedure and the filter criterion feel very heuristic and specific to the considered domains.
>
> **R3**: Yes, this actually indeed our goal -  providing a flexible and heuristic method applicable to various physical and planning domains involving numerous objects and interactions, in which object interactions are critical for effective planning and policy learning. Since the framework does not rely on specific domain physical priors or detailed domain information, it only requires object states, which can be obtained either from observations or off-the-shelf object-centric encoders. So the flexibility makes the framework can be applicable across many domains, rather than being limited to specific ones.
>
> Additionally, learning and leveraging interactions in physical domains is challenging, as many foundational models still struggle to accurately capture and understand physical interactions (see one recent evaluation [6]). The problem of identifying interactions from a causal perspective is related to the ongoing research problem of actual causality [7], with ongoing recent work demonstrating that this problem requires a computationally intractable search over the set of counterfactuals [8,9]. Therefore, we see the potential in this direction and believe in the scalability and utility of the null counterfactual framework.
>
> [6] Kang, Bingyi, et al. "How Far is Video Generation from World Model: A Physical Law Perspective." arXiv preprint arXiv:2411.02385 (2024).
>
> [7] Halpern, Joseph Y. Actual causality. MiT Press, 2016.
>
> [8] Chuck, Caleb, et al. "Automated Discovery of Functional Actual Causes in Complex Environments." arXiv preprint arXiv:2404.10883 (2024).
>
> [9] Beckers, Sander. "Causal explanations and XAI." Conference on causal learning and reasoning. PMLR, 2022.

---

> > ### Comment · Reviewer_EVkv · 2024-11-25
> > **Reviewer Response**
> >
> > I thank the Authors for their response.
> >
> > My overall concern is still that the proposed method feels very specific to the type of tasks being considered in the paper, and that many parts of the method seem very heuristic and hacky (rather than general and broadly applicable); to me, it seems difficult to apply those domain-specific heuristics to general RL problems, thus limiting the impact of this work in its current form.
> >
> > I will maintain my score.

---

> ### Author Response · Authors · 2024-11-26
> **More clarifications**
>
> We sincerely thank you for your feedback. We have provided a general response for all reviewers. Here, we would like to discuss your specific comments.
>
> First, we would like to clarify that the tasks discussed in our paper are not niche cases. Rather, learning and leveraging interactions are foundational and crucial research areas in general dynamic systems and probabilistic inference [1-2], and actual causality [3-5]. In RL, identifying and inferring object-centric dynamics and interactions is particularly important, as it helps agents better understand the structured dynamics and enhances decision-making efficiency [6-10]. As demonstrated in our paper, along with related works, incorporating interaction inference has shown positive effects on (GC)RL tasks in general.
>
> To further clarify our claim, our approach performs well in domains where counterfactual nulling can be applied, and where the distribution of goals is mismatched with the hindsight distribution. While these assumptions may not hold for every domain, they are relevant to several tasks such as robotic manipulation, logistics, and video games. In these domains, the states observed by an agent are significantly influenced by the policy the agent follows and the state factors (e.g., objects, facilities, game elements) it interacts with. Our empirical evidence shows that filtering the hindsight distribution using HInt provides clear benefits in sample efficiency.
>
> Additionally, we would like to emphasize that our model is not limited to physical interactions. Given the general nature of null counterfactual inference, our approach can be extended to interactions involving any state variables. Please refer to our previous responses for further clarification on this point.
>
> We hope this addresses your concerns. If you have any additional questions or points for discussion, we would be happy to further clarify. We truly appreciate your time, effort, and valuable suggestions to help improve our work.
>
>
> ---
>
> [1] Kipf, Thomas, et al. "Neural relational inference for interacting systems." ICML 2018.
>
>
> [2] Bleistein, Linus, et al. "Learning the dynamics of sparsely observed interacting systems." ICML 2023.
>
>
> [3] Halpern, Joseph Y. Actual causality. MiT Press, 2016.
>
> [4] Chuck, Caleb, et al. "Automated Discovery of Functional Actual Causes in Complex Environments." arXiv preprint arXiv:2404.10883 (2024).
>
> [5] Beckers, Sander. "Causal explanations and XAI." Conference on causal learning and reasoning. PMLR, 2022.
>
> [6] Wang, Zizhao, et al. "ELDEN: exploration via local dependencies." NeurIPS 2023.
>
> [7] Seitzer, Maximilian, Bernhard Schölkopf, and Georg Martius. "Causal influence detection for improving efficiency in reinforcement learning." NeurIPS 2021.
>
> [8] Yoon, Jaesik, et al. "An investigation into pre-training object-centric representations for reinforcement learning." ICML 2023.
>
> [9] Hwang, Inwoo, et al. "On discovery of local independence over continuous variables via neural contextual decomposition." ICML 2023.
>
> [10] Qiu, Yiwen, Yujia Zheng, and Kun Zhang. "Identifying Selections for Unsupervised Subtask Discovery." NeurIPS 2024.

---

### Official Review · Reviewer_rrvS · 2024-11-04

**Soundness:** 3
**Presentation:** 3
**Contribution:** 3
**Rating:** 6
**Confidence:** 4

**Summary:**

This paper proposes to leverage the null assumption to filter out states without interaction between the agent and object, improving the sample efficiency of GCRL. The approach begins by using a learned dynamics model to identify null states—where the next state remains the same in the absence of a specific state. It then keeps those trajectories where the agent directly interacts with the object, training the agent with hindsight relabeling. This approach shows comparable or superior sample efficiency in both 2D dynamic and 3D manipulation environments.

**Strengths:**

- The introduction of the *Null counterfactual interaction assumption* could be a important contribution, improving sample efficiency across various domains, particularly in manipulation tasks where interaction is minimal.
- The method details engineering practices to make the approach both manageable and efficient.
  + This includes null state inference with a dynamics model and predicting null operation.
- The paper presents a rich set of environments, and design choices for these environments, etc.

**Weaknesses:**

- **Scalability to high-dimensional state**
  + How is the state space defined across all environments? Assuming the entire environment has a low-dimensional state space, I’m curious how it computes the difference between states (Eq. 3) and infers the null state (Eq. 4) in a high-dimensional case (e.g., image).
  + From my understanding, inferring the null state should have a complexity of $O(n^2)$ based on state dimensionality, which may limit scalability in high-dimensional state spaces. However, L263 mentions a time complexity of $O(1)$. Could the authors clarify this?

- **Dependence on hyperparameters**
  + The method distinguishes null states based on prediction error (Eq. 3), but setting this hyperparameter could vary depending on environments and tasks.
  + Moreover, certain states, even within an environment or task, may have more complex dynamics than others. In such cases, how does the method define a single $\epsilon_{null}$?

**Questions:**

- While the authors use a learning-based dynamics model to infer the interaction, it can be clearly distinguished from existing work that utilizes other approaches. For example, [1] utilizes proprioceptive state changes to distinguish contact.
- The explanation of mixture distribution on L189 wasn't clear. How could it mix two distributions with a multiplication factor?
- The discussion on the limitation of this work can make readers better understand of the method. For example, authors can mention the domain where interaction is actually prohibitive (e.g., drone navigation)

#### Minor typo
- I believe L187 should be $d_{\pi}$

### References
[1] Manuelli and Tedrake, "Localizing external contact using proprioceptive sensors: The Contact Particle Filter"

---

> ### Author Response · Authors · 2024-11-19
> **Response to Reviewer rrvS (Part 1)**
>
> Thank you for your constructive comments. We have responded below.
>
> > Q1: How is the state space defined across all environments? Assuming the entire environment has a low-dimensional state space, I’m curious how it computes the difference between states (Eq. 3) and infers the null state (Eq. 4) in a high-dimensional case (e.g., image).
>
> **R1**: Thank you for pointing this out. For high-dimensional cases, we can address two scenarios:
>
> - Case 1: The object states are accessible but are high-dimensional. We believe our framework can still handle this scenario. To validate this, we are conducting evaluations on simulated high-dimensional random vectors and will provide results soon.
>
> - Case 2: The object states are not directly accessible, and only images/videos are available. In this case, we propose leveraging object- or slot-centric encoders to model the generative process from low-dimensional states to high-dimensional observations. The encoder's outputs can then serve as the state representations, allowing our framework to operate seamlessly. To test this, we are currently evaluating a combination of a VAE and object-state representation in the Box2D environment. We will share these results within this rebuttal period once they are available.
>
> > Q2: From my understanding, inferring the null state should have a complexity of $O(n^2)$
>  based on state dimensionality, which may limit scalability in high-dimensional state spaces. However, L263 mentions a time complexity of O(1). Could the authors clarify this?
>
> **R2**: This is because, during inference, we can directly use $h$, the learned inference model designed to align with the counterfactual test, to infer the relationship. Thus the process operates in O(1) time.
>
> > Q3: [Dependence on hyperparameters] The method distinguishes null states based on prediction error (Eq. 3), but setting this hyperparameter could vary depending on environments and tasks. Moreover, certain states, even within an environment or task, may have more complex dynamics than others. In such cases, how does the method define a single $\epsilon_\text{null}$?
>
> **R3**: We appreciate that the reviewer's identification of $\epsilon_\text{null}$ as a key parameter. In practice, we used the same null epsilon parameter of $\epsilon_\text{null} = 1$ for all environments and all experiments, even across domains such as random vectors, where the dynamics differ significantly from those in the physical interaction domains such as air hockey and SpriteWorld. This is because when the state inputs and deltas are normalized the effect of interactions as a result of an interaction is fairly significant. For example, the average change in velocity as a result of an interaction such as a ball hitting another ball or a robot manipulating an object in most physical domains is significant compared with the effect of drag. *We are also working on an ablation illustrating the effect of changing the $\epsilon_\text{null}$ value in both the random sprite domain and the random vectors domain (to show the difference across two significantly different dynamics), to illustrate the insensitivity of this hyperparameter, and will include that in this thread when those runs are completed.*
>
> We suggest the following strategy for selecting $\epsilon_\text{null}$: We can take the null model $f(\mathbf s, \mathbf a, \mathbb B(\mathbf v))$ and observe that the interaction states will be those where the difference between the nulled and non-nulled model outputs will be larger (meaning the non-nulled model will have higher likelihood.) On the other hand, on non-interaction states the error should be small. Thus, we can take the differences, identify two clusters, and then take the midpoint between the higher cluster center (corresponding to interaction states) and the lower cluster center (corresponding to non-interaction states. As $\epsilon_\text{null}$. **We add a formalization of this hyperparameter selection strategy in the appendix to strengthen the overall generalizability of the paper, and add some analysis illustrating the threshold selected by applying these operations.**
>
> *Related Revised Sections: Appendix E*

---

> > ### Author Response · Authors · 2024-11-19
> > **Response to Reviewer rrvS (Part 2)**
> >
> > > Q4: While the authors use a learning-based dynamics model to infer the interaction, it can be clearly distinguished from existing work that utilizes other approaches. For example, [1] utilizes proprioceptive state changes to distinguish contact.
> >
> > [1] Manuelli and Tedrake, "Localizing external contact using proprioceptive sensors: The Contact Particle Filter"
> >
> >
> >
> > **R4**: Thank you for the suggestion! We have added it as a contacting inference method in the revised related work section. However, our work, along with approaches like context-specific causal discovery, differs from these physical contacting inference models. Our focus is on identifying interactions using learning-based inference models from observational data. The null counterfactual inference does not assume access to physics priors (e.g., rigid body dynamics) of specific environments, which are more general and flexible.
> >
> > *Related revised section: Paragraph 1, Section 2*
> >
> > > Q5: The explanation of mixture distribution on L189 wasn't clear. How could it mix two distributions with a multiplication factor?
> >
> > **R5**: Apologies for the confusion. Here, we add the probability mass/density functions, scaling them appropriately to ensure they still sum/integrate to 1.  We added this point as a footnote in the revised version.
> >
> > *Related revised section: Section 3.1*
> >
> > > Q6: The discussion on the limitation of this work can make readers better understand of the method. For example, authors can mention the domain where interaction is actually prohibitive (e.g., drone navigation)
> >
> > **R6**: Thank you for the suggestions! We have incorporated these cases into the revised limitations section. We agree that in certain domains, such as locomotion, interactions may not play a critical role and can sometimes even be unhelpful or potentially harmful. However, in scenarios like driving or drone navigation, having a model that effectively captures interactions can also be useful, as it helps understand and avoid potential collisions or conflicts.
> >
> > *Related revised section: Section 6*
> >
> > > Q7: Minor typo
> >
> > **R7**: Nice catch, we have fixed it.
> >
> > *Related revised section: Section 3.1*

---

> > > ### Author Response · Authors · 2024-11-24
> > > **Update on high-dimensional states evaluation**
> > >
> > > Thank you once again for the insightful suggestions and comments. We have finished the experiments on high-dimensional states.
> > >
> > > These are the results for the misprediction rate (lower is better) of inference in additional domains from the state (similar to Table 1 in our original paper). Interactions are reweighted to constitute 50% of the test dataset. The boldface highlights the best result (approaching ~1) with the standard deviation. The k-in-nonlinear method incorporates nonlinearities into the random DAG instead of linear relationships (with 40-dim refers to a 40-dimensional state).
> > >
> > > | Method          | Nullname w/ Point   | JACI               | Gradient          | Attention         | NCD             |
> > > |-----------------|---------------------|--------------------|-------------------|-------------------|-----------------|
> > > | 1-in-nonlinear  | **0.9 ± 0.2**      | 2.4 ± 0.8         | 32.5 ± 6.1        | 37.4 ± 0.7        | 21.2 ± 1.1      |
> > > | 2-in-nonlinear  | **2.3 ± 0.1**      | **2.5 ± 0.2**     | 36.4 ± 0.4        | 21.8 ± 0.9        | 19.8 ± 2.0      |
> > > | 40-dim          | **1.4 ± 0.1**      | 2.4 ± 0.5         | 34.7 ± 4.4        | 26.4 ± 6.9        | 12.5 ± 0.8      |
> > >
> > >
> > > We will upload the revised version with these results, along with results from other ongoing experiments, once they are complete.

---

> > > > ### Comment · Reviewer_rrvS · 2024-11-25
> > > > **Concerns are well addressed. But high-dimensional state is still not clear.**
> > > >
> > > > Thank you to the authors for their detailed clarifications and the additional experiments provided. Several concerns, particularly those regarding hyperparameters, the significance of identifying interactions through a learning-based approach, and the limitations across different domains (e.g., navigation), have been well addressed. As a result, I have revised my score to "marginal accept."
> > > >
> > > > However, my primary concern regarding scalability to high-dimensional states remains unresolved. While leveraging an object-centric encoder could be a solution, learning the dynamics on top of this framework may not be straightforward.
> > > >
> > > > If the authors could provide a clear explanation or further insights on this matter, I would be willing to reconsider and potentially raise my score.

---

> ### Author Response · Authors · 2024-11-25
> **Updates on more high-dimensional states experiments and ablations on $\epsilon_\text{null}$**
>
> Thank you for acknowledging our response and raising the rating. We appreciate your continued discussion on high-dimensional cases. Here, we would like to provide an update with our new results on high-dimensional scenarios and discussions, along with ablation studies on $\epsilon_\text{null}$.
>
> ### **High-dimensional cases**
>
> - NCII using VAE encodings table:
>
> See *General Response, Appendix J*
>
> - HInt using VAE：
>
> See *General Response, Appendix J*
>
> We provide some preliminary experimental results that verify that NCII and HInt have the potential to scale to higher-dimensional domains. In these experiments, we use state-segmented 80x80 object masks for each object. The input state is then pixel-based statistics on the objects (position, delta position, etc.), as well as the latent state of a 128-latent dimension variational autoencoder trained to encode all of the segmented objects. The pixel-based statistics are primarily to encode dynamic information like velocity, which is often a small (if any) portion of the variational autoencoder. To match the encoding dimension of the latent space, the pixel statistics are tiled to 128 dimensions. Together, each factored state is a 256-dimensional input and is passed into NCI, and used as the observation for RL using hindsight filtering. Additional details can be found in **Appendix J (Figure 11, Table 10)**.
>
> The performance of all existing interaction-based methods, and the performance of GCRL, decline in the image-based domain and noise increase. This is unfortunately also the case with NCII and HInt. **Nonetheless, by comparison to the baseline methods, both methods show improved performance**. Furthermore, a VAE-based encoding may not be ideal for NCII-based methods. Future work might investigate object-based representation [1-3], and image-based goal-conditioned methods [4,5]. Investigations of this sort, while important, are tangential to the research objective of this work: to demonstrate that the null hypothesis can be a valuable inductive bias for interaction inference and that using interactions for hindsight filtering can significantly improve the performance of goal-conditioned RL. We suggest that the existing experiments in the paper demonstrate this clearly and that the additional experiments exploring higher dimensional states validate this claim. Incorporating the elements investigated through NCII and HInt would undoubtedly be invaluable for future work in scaling GCRL and interaction inference, but both of these general research areas remain open problems when scaling to more complex domains.
>
> [1] Yoon, Jaesik, et al. "An investigation into pre-training object-centric representations for reinforcement learning." ICML 2023.
>
> [2] Zadaianchuk, Andrii, Maximilian Seitzer, and Georg Martius. "Object-centric learning for real-world videos by predicting temporal feature similarities." Advances in Neural Information Processing Systems 36 (2023).
>
> [3] Li, Yulong, and Deepak Pathak. "Object-Aware Gaussian Splatting for Robotic Manipulation." ICRA 2024 Workshop on 3D Visual Representations for Robot Manipulation.
>
> [4] Chane-Sane, Elliot, Cordelia Schmid, and Ivan Laptev. "Goal-conditioned reinforcement learning with imagined subgoals." International conference on machine learning. PMLR, 2021.
>
> [5] Zhang, Zichen, et al. "Universal visual decomposer: Long-horizon manipulation made easy." 2024 IEEE International Conference on Robotics and Automation (ICRA). IEEE, 2024.
>
> ----
>
> ### **Update on $\epsilon_\text{null}$**
>
> See *General response, Table 2 in Appendix E*
>
>
> We ablated on the $\epsilon_\text{null}$ hyperparameter with 3 seeds for each setting of $\epsilon_\text{null}$ to empirically analyze the dependence on the threshold for identifying null interactions. Our experiments evidence a limited dependence on this parameter except at the upper and lower extremes, as values within 0.3 to 2.5 show performance within variance for both random DAG and Box2D domains, which have significantly different dynamics. Since epsilon-null compares the difference in normalized log-likelihood of the predictions, this is an invariance across two orders of magnitude. This result corroborates with the evidence that across all of the domains that we tested, air hockey, Spriteworld, Robot pushing, Franka Kitchen and random DAG, and Random DAG with nonlinear relationships, the same $\epsilon_\text{null} = 1$. We believe that this suggests the efficacy of NCII across a variety of domains, though future work can investigate strategies for automatically setting the $\epsilon_\text{null}$ parameter.
>
> Please feel free to let us know if you have any further questions. Thank you again for your valuable comments and efforts!

---

### Author Response · Authors · 2024-11-25
**General Response (1/2)**

We sincerely thank all reviewers for their time and valuable feedback. We greatly appreciate their recognition of our work's ideas and contributions as being "important" (```rrvS```), a "promising direction" (```EVkv```), and "well motivated" (```xepm```, ```SPH4```); the methodology as "manageable and efficient" (```rrvS```), "an important technique in goal-conditioned RL" (```xepm```), with experiments demonstrating "comparable or superior sample efficiency" (```rrvS```), a "rich set of environments" (```rrvS```), and "practical gains in object-centric robotic domains" (```EVkv```), "relevant, sufficiently established, and significant benefits" (```SPH4```). We are also grateful for their positive comments on the presentation, describing it as "good" (```rrvS```, ```EVkv```, ```xepm```), "smooth and well thought out" (```xepm```), with an "intuitive and concise interpretation" (```SPH4```), and "well structured" (```SPH4```).


In this general response, we summarize our replies to the common concerns, along with the major revisions and new experiments we have conducted.


### **Common concerns**


- Dependency on Hyper-parameter $\epsilon_\text{null}$ (```rrvS```, ```xepm```)


We thank you for your pointer regarding this. We have addressed in the related responses (**R3** to ```rrvS```, **R1** to ```xepm```).


In our experiments, we used the same null epsilon parameter across all environments, including those with significantly different dynamics, such as random vectors and physical interaction domains like air hockey and SpriteWorld. This was because, when state inputs and deltas are normalized, the impact of interactions tends to be significant. We also conducted an ablation study to explore the impact of varying this parameter (**Table 2** in Appendix). Additionally, we have provided the strategies for identifying this parameter using information
from learning the null model (see **Appendix E**).

- Scalability (```rrvS```, ```EVkv```)


Thank you for pointing this out. First, we would like to clarify that the null counterfactual inference proposed here is not restricted to physical interactions due to its general conceptual nature. Additionally, it is not necessarily limited to use with object-factored states. The framework is also architecture-agnostic, meaning it can be integrated with various deep learning models such as PointNet, GNNs, and transformers, as demonstrated in the paper. Hence, we believe this approach is scalable, as evidenced by its empirical usefulness in goal-conditioned RL domains. Furthermore, it has the potential to be combined with larger object-centric pre-trained perception models (though this is not the focus of our work) to enhance scalability for real-world data. The detailed discussion can be seen in the responses (**R1** to ```rrvS```, **R2** to ```EVkv```).


We also want to note that the claim of this work is that in domains where counterfactual nulling can be used, and where the distribution of goals is mismatched with the hindsight distribution, our method can perform well. These assumptions, while they do not apply to every domain, apply quite broadly to tasks such as robotic manipulation and logistics, or video games. In all of these settings, the states that might be observed by an agent will be significantly impacted by the policy the agent takes and the state factors (objects, facilities, game elements) it interacts with. In this work, we provide empirical evidence that filtering the hindsight distribution using HInt provides a clear sample efficiency benefit.


Finally, we would like to point to the high-dimensional state results we obtained during the rebuttal process suggesting that HInt can also be scaled to image inputs. Please refer to the revised **Appendix J** for details.


- For the other points raised, such as typos, suggested changes to certain equations, and updates to the appendix, we have addressed all of them in the revision. Please refer to our updated manuscript and point-by-point responses for details.

---

> ### Author Response · Authors · 2024-11-25
> **General Response (2/2)**
>
> ### **More Evaluations and Major Modifications in the Revision**
>
>
> **[Regarding evaluating scalability in high-dimensional cases]**
>
>
> - NCII on nonlinear and high dimensional state compared to baselines (**Table 9**)
> - NCII on the image-encoded state (Appendix J, **Table 10**)
> - HInt with Image encoded state as input (Appendix J, **Figure 11**)
>
>
>
>
> **[More ablation studies]**
>
>
> - NCII with different $\epsilon_\text{null}$ (Appendix E, **Table 2**)
> - comparing different kinds of hindsight (final, episode, and future). (Appendix I, **Figure 10**)
> - wall clock time comparison (**Table 8**)
>
>
> **[More baselines]**
>
>
> - CAI baseline (**Figure 12**)
>
> Other clarifications are highlighted in red throughout the revision. We hope that our responses have addressed the reviewer’s concerns and remain available for any follow-up questions. Thank you again for your time and effort!

---

### Meta-Review · Area_Chair_U5Ve · 2024-12-21

**Metareview:**

The paper proposes a novel variant of HER, dubbed Hindsight relabeling using Interactions (HInt). HInt leverages the concept of Null Counterfactual Interaction Inference to improve sample efficiency in goal-conditioned reinforcement learning tasks in object-centric robotic environments. The method filters trajectories based on detected interactions between objects, aiming to focus learning on those where the agent's action affects the target object. Empirical evaluations demonstrate improved sample efficiency compared to baseline methods.

Reasons to accept
- The introduction of null counterfactual interactions offers a novel inductive bias that enhances the relevance of trajectories in HER, potentially improving sample efficiency.
- The method’s foundation in causality, where an interaction is defined as an influence of one object on another’s transition dynamics, is both intuitive and compelling.
- The paper provides strong empirical results showing that the proposed method outperforms established methods like HER, especially in environments where object interactions are key to the agent's task.
- The problem setup is well-motivated, and the connection to HER is well-established, making the contributions accessible to the audience.

Reasons to reject
- The filtering method seems to rely heavily on heuristics and domain-specific knowledge, which may limit the method's generality, particularly in non-object-centric domains.
- The method depends on several hyperparameters, such as the threshold for interaction detection, which could vary across environments. A more thorough ablation study on hyperparameter sensitivity is needed to understand the robustness of the approach.
- While the method shows improvements over HER, comparisons with other relevant approaches, such as CAI (Causal Influence Detection), are missing, which would help validate the generality and superiority of the proposed method.

During the author-reviewer discussion phase, many of the reviewers' concerns were well-addressed, e.g., adding the CAI baseline. During the AC-reviewer discussion phase, the main focus comes down to the method's generality, as the proposed approach seems specific to object-centric tasks with relatively simple object-centric dynamics, and it seems difficult to adapt this approach in its current form to real-world robot tasks. Yet, in my opinion, since ICLR is a machine learning venue, instead of a robotic one, we should put more emphasis on algorithmic aspects and contributions of this approach and how it can inspire future works along the line. Consequently, I recommend accepting the paper.

**Additional Comments On Reviewer Discussion:**

During the rebuttal period, all four reviewers acknowledged the author's rebuttal, and two reviewers adjusted the score accordingly.

---

### Decision · Program_Chairs · 2025-01-22

Accept (Poster)